# Gamma-ray Emission and Variability Processes in High-Energy-Peaked BL Lacertae Objects

**Bidzina Kapanadze [1,2,3]**

1 Space Research Center, Department of Astronomy and Astrophysics, School of Natural Sciences and Medicine, Ilia State University, Cholokashvili Av. 3/5, Tbilisi 0162, Georgia; bidzina_kapanadze@iliauni.edu.ge or bkapanadze@abao.ge; Tel.: +995-577413299

2 E. Kharadze National Astrophysical Observatory, Deapartment of Galaxies and Stars, Mt. Kanobili, Abastumani 0803, Georgia

3 Instituto National di Astrofizica e Fizica spatiale, Osservatorio Astronomico di Brera, Via E. Bianchi 46, 23807 Merate, Italy

**Abstract:** BL Lac objects are active galactic nuclei notable for a beamed nonthermal radiation, which is generated in one of the relativistic jets forming a small angle to the observer's line-of-sight. The broadband spectra of BL Lacs show a two-component spectral energy distribution (SED). High-energy-peaked BL Lacs (HBLs) exhibit their lower-energy (synchrotron) peaks at UV to X-ray frequencies. The origin of the higher-energy SED component, representing the $\gamma$-ray range in HBLs, is still controversial and different emission scenarios (one- and multi-zone synchrotron self-Compton, hadronic etc.) are proposed. In $\gamma$-rays, HBLs show a complex flaring behavior with rapid and large-amplitude TeV-band variations on timescales down to a few minutes. This review presents a detailed characterization of the hypothetical emission mechanisms which could contribute to the $\gamma$-ray emission, their application to the nearby TeV-detected HBLs, successes in the broadband SED modeling and difficulties in the interpretation of the observational data. I also overview the unstable processes to be responsible for the observed $\gamma$-ray variability and particle energization up to millions of Lorentz factors (relativistic shocks, magnetic reconnection, turbulence and jet-star interaction). Finally, the future prospects for solving the persisting problems by means of the dedicated gamma-ray observations and sophisticated simulations are also addressed.

**Keywords:** galaxies; BL Lacertae objects; general

## 1. Introduction

Blazars are active galactic nuclei (AGNs) which are commonly understood as having the relativistic jets emerging from the central supermassive black holes (SMBH; $M \sim 10^8$–$10^{10}$) and forming small angles with respect of our line-of-sight ($\theta < 10$–15 deg). Consequently, the relativistic motion of the plasma boosts the non-thermal jet emission into a forward cone pointed to the observer [1]. Owing to such a favourable geometry, the strongly beamed jet radiation often completely outshines the other AGN components [2]. Accreting SMBHs are believed to convert their rotational energy into Poynting flux and power the collimated jets (see [3] and references therein).

BL Lacertae objects (BL Lacs) are a blazar subclass which demonstrate featureless spectra [1] and represent a majority of the AGN detected so far in the TeV band[1] (56 out of the total 89). Their broadband SED consists of two smooth, broad distinct components [4]: the first "hump" extends from the infrared to X-rays (synchrotron emission from relativistic electrons residing the jet emission zone), and a high-energy component having a peak at the MeV–TeV energies. The origin of the latter is still under debate and three fundamentally different approaches have been proposed: *leptonic*, *hadronic* and hybrid *lepto-hadronic* models, based on the particles responsible for the $\gamma$-ray emission (see, e.g., [5]).

Depending on the position of the synchrotron SED component, BL Lacs are broadly divided into the low-energy-peaked (LBLs) and high-energy-peaked (HBLs) objects; one additionally considers a group of intermediate-energy-peaked BL Lacs (IBLs; see [4,6]). The sequence LBLs→IBLs→HBLs is characterized by increasing peak frequency in the $\nu$–$\nu F_\nu$ plane, declining dominance of the $\gamma$-ray flux over the lower-energy emission, and decreasing bolometric luminosity. In the HBL objects, the lower-energy peak is situated at UV-to-hard X-ray wavelengths; their high-frequency component peak is generally situated beyond $\sim$100 GeV, and the first SED peak is up to one order higher than the higher-energy one (see Figure 1 and [7]). Moreover, the subclass is widely accepted to possess radiatively inefficient accretion disks (see, e.g., [8]). Note that among the extragalactic TeV sources, the highest energy photons (up to 20 TeV) have been reported for the HBL source Mrk 501 [9].

Generally, the HBL spectra observed above 300 GeV by imaging atmospheric Cherenkov telescopes (IACTs) are frequently quite steep (with the photon index $\Gamma \gtrsim 2.5$), defining an SED turnover [10]. Moreover, very-high-energy $\gamma$-rays (VHE, $E > 100$ GeV) emitted by the objects situated beyond $\gtrsim$100 Mpc reach us impacted by significant absorption caused by the extragalactic background light (EBL; via the process $\gamma\gamma \to e^- e^+$). Namely, the $\gamma$-ray spectrum undergoes a strong deformation at energies characterized by the optical depth $\tau(E, Z) \gtrsim 1$ [11]. Various studies have revealed that EBL contains two components, namely, at the near- and far-infrared wavelengths, separated by a mid-infrared (MIR) "valley" [12]. Consequently, $\tau(E)$ was found to be strongly dependent of the photon energy below 1 TeV and above 10 TeV, while this dependence is much weaker between 1 and 10 TeV. Therefore, one expects a significant distortion of the VHE spectra of HBLs at energies below 1 TeV and above 10 TeV [11]. Note that the range of 100 MeV–100 GeV covered by the Large Area Telescope (LAT) on board *Fermi* (*Fermi*-LAT; [13]) is characterized only by small $\gamma\gamma$ -attenuation and negligible below 3 GeV [10]. This allow us to discern the underlying particle population more robustly.

Among HBLs, one can additionally discern (i) extreme high-energy peaked BL Lacs (EHBLs), with the synchrotron SED peak $E_{\mathrm{p}}^{sync} \geq 1$ keV and showing a hard X-ray photon index $\Gamma_{\mathrm{x}} < 2$ [14]; (ii) five TeV-detected objects ( 1ES 0229+200, 1ES 0347−121, 1ES 1101−232, 1ES 1218+304 and RGB J0710+591) exhibit a higher-energy peak $E_{\mathrm{p}}^{\gamma} \geq 1$ TeV and a hard photon index $\Gamma_{\gamma} \leq 1.5$–1.9 in the 0.1–10 TeV band, after correction for the EBL [15]. Such sources were classified as ultra-high-energy peaked BL Lacs (UHBLs, see [16]), as well as being EHBLs. The discovery of these UHBLs was a surprise, since the standard emission models yield the higher-energy peak below 1 TeV for HBLs due to the limited maximum energy of the electrons and the Klein-Nishina (KN) effects for X-rays, strongly suppressing the TeV-band emission [8].

This review is focused on the results achieved by the different emission scenarios attempting to explain the origin of $\gamma$-ray emission in HBLs. They provide us with an efficient tool to evaluate the physical parameters describing the jet emission zone (by comparing the observed higher-energy SED with those modelled in the framework of the different emission scenarios) and draw conclusions about the jet particle content. In turn, information from the VHE part of the SED is required to constrain model parameters for HBLs, which radiate a significant part of their overall $\gamma$-ray emission in that energy range. We briefly review also those acceleration processes which are primary "candidates" for energizing the jet particles up to ultrarelativistic energies required for producing $\gamma$-ray photons either by inverse Compton (IC) upscatter or hadronic mechanisms (shocks, magnetic reconnection, magnetohydrodynamic turbulence and magnetospheric vacuum gap), as well as represent the sources of the observed variability on various timescales.

One can not directly resolve the HBL emission zone due to its extremely small angular size. Therefore, a multiwavelength (MWL) variability study represents practically the only way for drawing conclusions about the structure of the jet emission zone. Especially informative is the $\gamma$-ray variability study since this emission is produced by the highest energy electrons, which lose energy very quickly and exist only in the vicinity of the acceleration sites. Nevertheless, the VHE emission of HBLs are characterized by the most

rapid variability (since the cooling time at these energies are the shortest) and based on light travel time arguments, the corresponding timescales impose constraints on the size of the emission region. The most challenging is an ultra-fast variability shown by some close, bright HBLs on timescales down to a few minutes [17,18]). The latter are significantly shorter than the light-crossing time of the central SMBH and one requires requires extreme physical conditions for their interpretation [2]. Consequently, the $\gamma$-ray variability allows us to discern the physical processes operating in the innermost jet area and also represent one of the subjects of the current review.

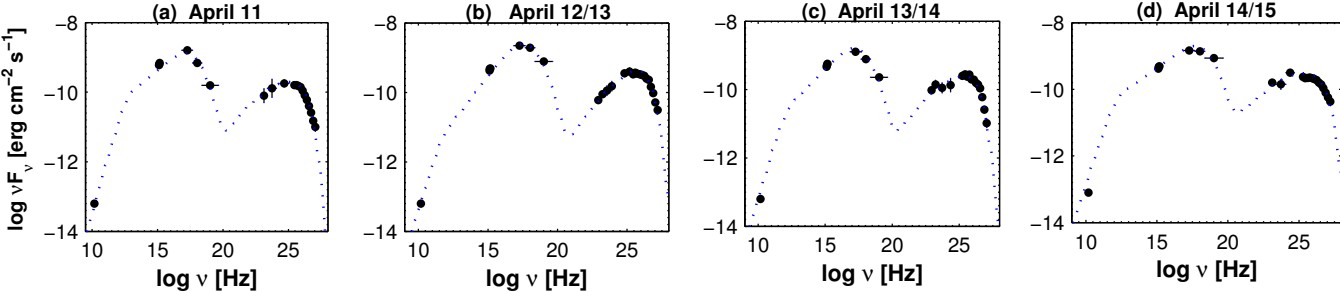

**Figure 1.** Broadband SEDs of the HBL source Mrk 421 where Panels (**a**–**d**) correspond to the different nights in the epoch of the exceptional X-ray flare in 2013 April. The dashed blue lines represents the one-zone SSC model. Reproduced by permission of AAS from [19].

First of all, we present a review of the $\gamma$-ray emission processes in HBLs, their advantages and limitations (Section 2). The $\gamma$-ray variability and underlying unstable processes are discussed in Section 3. Finally, we discuss the future prospects of $\gamma$-ray observations and the associated simulations for HBLs.

## 2. Emission Mechanisms

In leptonic models, $\gamma$-ray emission is produced by the jet leptonic content (electrons/positrons), while protons do not possess sufficient energies for the photo-pion generation process and significant proton-synchrotron radiation (or such high-energy protons could exist, but not in a high enough number to dominate the electromagnetic emission; see, e.g., [5,20]). The $\gamma$-ray emission of HBLs can be produced by the IC upscattering of low-energy photons by the "parent" electron population (*synchrotron self-Compton* model, SSC; [21] and references therein), or "seed" photons can be of external origin (so-called *external inverse-Compton* model, EIC; [7,22]). However, the entire spectrum is likely due to a combination of direct synchrotron and SSC emission in HBLs, without any significant component due to the upscattering of externally produced photons: these sources do not exhibit any significant external radiation fields from the disk, the broad line region (BLR), or the dust torus (e.g., [23]). The EIC process on photons from different parts of the HBL jet seems to be the only possibility [24]. Alternatively, hadronic or lepto-hadronic scenarios have also been considered as the gamma-ray emission mechanism to solve the difficulties with leptonic models [5].

### 2.1. One-Zone SSC Model

In the framework of the standard SSC scenario, the higher energy gamma-rays detected from HBLs arise from the IC upscattering of radio-to-X-ray photons by the "parent" ultra-relativistic electrons ([22] and references therein) accelerated in a jet which itself moves at relativistic speeds [25]. If external photon fields are neglected (as widely accepted for HBLs), the stationary single-zone SSC model can describe the steady MWL emission. Generally, the SSC models require low magnetic fields for HBLs (0.01–0.1 Gauss), which are significantly different from the equipartition between the magnetic and kinetic energy densities in the $\gamma$-ray emitting zone [5].

Within the *homogeneous one-zone* model, the emission zone is generally represented by a spherical "blob" containing a homogeneous magnetic field and a single lepton population. The latter may have five different functional shapes as follows (see, e.g., [20,26]): (1–2) simple and broken power laws; (3) logparabolic; (4) power law with exponential high-energy cutoff and (5) power law at low energies with a log-parabolic high-energy branch. The region moves with constant relativistic velocity $\beta_\Gamma c$ (of the bulk Lorentz factor $\Gamma = 1/\sqrt{1-(V/c)^2}$ with $V$, the bulk speed) towards the observer, forming a small viewing angle $\theta$. This leads to the boost in the recorded emission by the Doppler factor $\delta = 1/\Gamma(1-\beta cos\theta)$. The electron energy distribution (EED) cools through the synchrotron and IC mechanisms. One considers a temporary equilibrium between particle injection/acceleration, radiative cooling, and escape from a spherical emission region (see, e.g, [5,20] for the corresponding reviews), while an adiabatic expansion of the blob is considered in the one-zone expanding leptonic model discussed below. Generally, the EEDs described by the aforementioned functions can be derived by means of the Fokker–Planck equation, which incorporates the terms corresponding to particle acceleration, radiative (and, possibly, adiabatic) cooling, and particle escape (see, e.g., [27]).

Within the standard one-zone SSC models, the particle spectral index ($\sigma$) can be related the photon index ($\alpha$) as $\sigma = 2\alpha - 1$, if the EED is characterized by insignificant radiation cooling [28]. If there is a strong cooling via the IC-upscattering or the synchrotron mechanism acceleration of the leptons accelerated at the relativistic shock front (see Section 2.1), followed by an escape into the emission zone situated at the shock downstream region, the time-averaged effective EED is given by $\sigma = 2\alpha - 2$ [28].

The position of the IC component peak depends on the upscattering regime. If the electron energy is below $m_e c^2$ in the center-of-momentum frame, the electrons will be non-relativistic and the upscatter is characterized by the Thomson cross-section $\sigma_T$ (so-called *Thomson limit*; see, e.g., [29]). In the case the particle energy is higher than $m_e c^2$, the KN limit applies and the upscatter cross-section declines with increasing energy. Namely, the Thomson cross-section in the *head-on* approximation $\sigma_T = (8\pi/3)r_e^2$, with $r_e^2$ to be the classical electron radius. In the KN limit, however, $\sigma = (3/8)\sigma_T \epsilon^{-1}\ln(1+2ln2\epsilon)$, with $\epsilon = h\nu/m_e c^2$ [30]. In the laboratory frame, $\epsilon \ll \gamma^{-1}$ in the Thomson regime and the upscattered photon energy $\epsilon_s \approx \gamma^2\epsilon$, while the KN regime yields $\epsilon_s \approx (1/2)\gamma\epsilon$ with $\epsilon \gg \gamma^{-1}$. In the Thomson regime, the ratio of the SSC peak frequency to the synchrotron 'counterpart' $\nu_{ssc}^p/\nu_{syn}^p = (4/3)\gamma_p^2$, with $\gamma_p$, the EED peak [30]. In the case of the KN-upscattering ($\gamma_p h\nu_{syn}^p \gtrsim m_e c^2$) and the SSC peak frequency is given by $\nu_{ssc}^p \approx (2/\sqrt{3})(\gamma_p m_e c^2/h)$. The reduction of the cross section in the KN-regime significantly decreases the IC-upscattering efficiency [20].

In the SSC interpretation, the separation of the $\gamma$-ray emission zone from the central SMBH is not strictly constrained [31]. However, there is a constraint on the $\gamma$-ray luminosity from the pair opacity whose rate depends on the energy density of the produced radiation in the jet rest frame. This process is characterized by the maximum cross-section $\sigma = 3\sigma_T/16$ when there is a collision between the $\gamma$-rays photon of energy $\epsilon$ and the target photons with $\epsilon_t = 1/\epsilon$. The minimum energy threshold for this process is $\epsilon_0 = 0.26(E/TeV)^{-1}$ in the case of the head-on collision ($\theta = \pi$). Generally, the TeV opacity is primarily determined by the infrared photons [32]. Therefore, intense infrared fields prevent $\gamma$-rays to escape from the emission zone. Moreover, the EBL limits the redshift of HBLs (similar to the other AGN subclasses) from which their TeV-band photons can reach the Earth [32,33]. For the spherical and isotropically emitting jet area, the optical depth of the absorption process is related to the TeV-band luminosity as $\tau_{\gamma\gamma} = (3\sigma_T/16)(L(\epsilon_t)/(4\pi mc^3 R)$. Therefore, the condition $\tau_{\gamma\gamma} \lesssim 1$ requires that the TeV-band luminosity to be constrained as ($L(\epsilon_t \lesssim) \times 10^{43} M_8 R/r_g$ erg s$^{-1}$, with $M_8$ to be the central SMBH mass in units of $10^8 M_\odot$; $r_g$, the gravitational radius of the central SMBH. Frequently, the observed isotropic TeV-band luminosity of HBLs is one or two orders of magnitude higher compared to this limit (see, e.g., [32]). In combination with the short variability timescales observed for HBLs at the TeV frequencies, this result implies that this emission is strongly Doppler boosted and,

correspondingly, generated in a relativistic jet having a large bulk Lorenz factor $\Gamma$; a very short timescale implies a compact emission region (see Section 3) which, in turn, can be characterized by a large pair opacity, and the TeV emission could not escape from the jet otherwise.

In turn, this process restricts the location of the $\gamma$-ray emission region detected by us [32,34]; it is possible that the synchrotron flares generated at smaller radii than the surface $\tau_{\gamma\gamma}$=1 ("the $\gamma$-sphere"; [35]) will have no TeV counterpart. For example, the innermost AGN area up to 40 gravitational radii from the central SMBH was found to be opaque for the TeV-band emission in PKS 2155–304 [32]. However, this opacity is significantly reduced if the disk represents a radiatively inefficient accretion flow (RIAF; see [32,36]).

$\gamma$-rays from HBLs may pass through the massive stellar cluster surrounding the jet and luminous stars can emerge close to our line-of-sight. Consequently, the soft radiation field of these stars can absorb the jet-emitted $\gamma$-rays. Ref. [37] showed that this process is capable of producing a broad spectral dip in the range of 50–200 GeV, and the time scale of this event was found to a few to tens of days. On the other hand, the jet-surrounding red giants may introduce some large wind-blown "bubbles" into the jet, produce a double-shock structure there. Consequently, the jet particles can be accelerated to (ultra)relativistic energies and contributing to the detected $\gamma$-ray emission (see Section 2.4).

The homogeneous one-zone leptonic models were successful in explaining the SEDs and correlated variability in different HBLs. For example,

- Ref. [38] reported strong variations in both X-ray and TeV bands from the MWL observations of Mrk 421 in 1998 April, which were highly correlated and compatible with the standard one-zone SSC model. Similar results were obtained from the MWL campaigns performed in March 2001 [39], January 2006–June 2008 [40], 2009–2012 [41,42], March 2010 [43], January–June 2013 [19], December 2015–April 2018 [44] and for the VHE flares detected with FACT during December 2012–April 2018 [45].

- Ref. [46] modeled the TeV-band variability of Mrk 501 during the MWL campaign in 1994 within the homogeneous SSC model by fitting the quiescent spectrum of the source and then changing the maximum energy of the electron injection spectrum. This produced changes only in the X-ray and TeV bands, leaving all the other bands essentially unaffected. Ref. [47] modeled the April–May 1997 outburst in Mrk 501 by means of the time-dependent SSC model: a steady X-ray emission was combined with a variable SSC component and, moreover, a pre-acceleration of electrons up to $\gamma_{min} = 10^5$ was also assumed. The follow-up MWL flare in June 1998 was also modeled by means of one-zone SSC scenario, involving a significant increase in the magnetic field strength and in the electron energy by factors of 3 and 10, respectively [48]. Ref. [49] identified individual TeV and X-ray flares and found a sub-day lag between them (consistent with one-zone SSC model) during the FACT monitoring of the source in 2012 December–2018 April. Mrk 501 showed a low activity during the MWL campaign in 2008 March–May and the one-zone SSC model adequately described the broadband SED [50]. Similarly, the 0.3–10 keV flux was correlated with the HE and VHE emissions during 2017–2020 when the source showed the lowest historical X-ray and $\gamma$-ray states [51]. The average SED of Mrk 501 constructed via the data obtained during the MWL campaign performed in March–August 2009, successfully described within the one-zone SSC model with the dominant emission region characterized by the size smaller than 0.1 pc. The total jet power constituted only a very small portion ($\sim 10^{-3}$) of the Eddington luminosity and broken power-law EED was adopted [52].

- Ref. [53] adopted the one-zone SSC model for the broadband SEDs 1ES 1959+650 from the MWL campaign performed in 2012 April–June and deduced that the physical parameters describing the emission zone during the flaring states are significantly different from those corresponding to the low states. The MWL SEDs from the time window 13–14 June 2016 were modeled with the one-zone SSC scenario, requiring relatively large Doppler factors $\delta$ = 30–60 [54].

- PKS 2155–304 showed an active $\gamma$-ray flaring phase in 1997 November with a similar behavior in X-rays, compatible with the one-zone SSC scenario [55].

The SED "standard" HBLs with steep VHE photon inices ($\Gamma_{\text{VHE}} > 2$) can be modeled within the standard one-zone SSC scenario, using only a few physical parameters (redshift, radius of the emission region, magnetic field strength and Doppler factor (see [5,15]). However, to achieve a satisfactory description of UHBL sources within this model, one requires two essential ingredients: (1) even lower magnetic fields compared HBLs ($\lesssim 10\,\text{mG}$). This is required to (1) avoid a softening of the $\gamma$-ray spectrum by synchrotron cooling of the ultrarelativistic electrons; (2) explain a large separation between the synchrotron and SSC peaks; (2) a large minimum energy and peculiar EED dominated leptons of large Lorentz factors ($\overline{\gamma_e} \sim 10^3$–$10^4$; [15]).

Note that the photon–photon absorption process can yield the arbitrarily hard spectra by assuming that the $\gamma$-ray emission passes through the medium containing a hot photon gas with a narrow energy distribution characterized by $E_\gamma \epsilon_0 \gg m_e c^2$. In such a situation, the medium becomes optically thick at the lower $\gamma$-ray energies and thin at a higher one (due to the decrease in the cross-section of the $\gamma\gamma$ interaction). Consequently, the formation of the intrinsically hard $\gamma$-ray spectra can be achieved [11].

While the steady-state lepton models can be used to broadly characterize different activity states of HBLs, a time-dependent description of the electron distribution and/or of the source parameters is required in order to model fastly variable emission: flares can then be modelled through an interplay of particle injection or acceleration with particle cooling and escape, following, e.g., the Fokker–Planck equation as a function of time, or through sharp changes in the magnetic field, Doppler factor or physical extension of the emission region [56].

The main drawback of one-zone SSC models is an assumption that the highest-energy variability of the synchrotron and SSC emissions is produced by the most energetic electrons, the cooling timescale of which is significantly shorter than the light-crossing time of the emission zone. Even if any disturbance in the radiating medium instantaneously passes the emission zone, the observed emission will not contain the information about the fluctuations occurring on timescales shorter than the light-crossing time: they are smeared out owing to light-travel time delays from different source parts [57]. Moreover, representation of the synchrotron component with a single power-law electron distribution is frequently unsuccessful, and broken power-law EED with a much steeper second slope or a logparabolic model are required [5]. This situation indicates that the basic homogeneous one-zone SSC scenario is a simplification and, consequently, a rather complex representation of particle populations and the presence of different inhomogeneities in the emission region are required, in combination with the effects related to the particle injection, acceleration, escape and cooling processes [57]. Moreover, one should account for the differences between the physical conditions within and outside of the emission region (magnetic field strength, particle density etc.).

Moreover, a number of the MWL campaigns challenged the homogeneous one-zone SSC model. For example,

- In June 2004, 1ES 1959+650, underwent a strong "orphan" TeV flare by more than 4 Crab and 7 hr of doubling timescale without simultaneous X-ray event [24,58]. Similarly, strong $\gamma$-ray flares in 2009 May and 2012 May were not accompanied by those at synchrotron frequencies. In turn, no significant $\gamma$-ray activity was observed during some X-ray flares [59]. A similar behavior was also evident during 2006–2008 [60], January 2016–November 2017 [61–63]. Such events are very difficult to explain within the standard one-zone standard SSC scenarios.
- Mrk 421 underwent a very strong X-ray flare by a factor of 7 within 3 days during the MWL campaign in December 2002–January 2003, which was not accompanied by a comparable TeV-band activity [64]. During the giant flare in 2004, the TeV-band brightness reached its peak several days earlier the X-ray one that was inconsistent with the standard one-zone SSC model, and [65] suggested to be an instance of an

"orphan" TeV flare. [40,66] also found some high X-ray states, not accompanied by TeV flaring and vice versa in 2005–2008. Similar instances were reported by [42,44,67,68] from the periods February 2010–March 2013, November 2015–June 2015 and December 2015–April 2018, respectively. Moreover, there was a quadratic relation between X-ray and VHE variabilities during both the rising and decaying phases of a flare [39]. This is not expected in the KN regime [24]: the $\gamma$-ray emission is produced by the electrons having TeV and higher energies, which do not upscatter self-produced synchrotron photons since this is not possible owing to the smaller cross-section typical to the KN regime. However, such particles are capable for upscattering the lower-energy photons (produced by lower-energy electrons) in the Thomson regime. Consequently, the two peaks of the HBL SED are not produced by electrons having the same energy. Consequently, the VHE emission is expected to track the X-ray variability only linearly (instead of quadratically, as shown by Mrk 421). Particularly challenging is to observe a quadratic X-ray–TeV relation in the flare declining phase, owing to the similar energy dependence of both synchrotron and IC cooling ($\propto \gamma^2$) and again, a linear dependence is expected. A quadratic decrease can be achieved even in the Thomson regime, although extremely large beaming factors are required [24,39]. Nevertheless, Mk 421 showed even a super-quadratical X-ray–VHE relation during the fast flare on 19 March 2001 [39].

- During the exceptionally strong X-ray outburst of Mrk 501 in 2014 March–October, the 0.3–10 keV flux was generally correlated with the TeV-band emission, while there was no significant correlation between the 0.3–300 GeV and optical–UV flux variations. Moreover, several cases of the complicated X-ray and $\gamma$-ray variabilities were reported, which were inconsistent with the one-zone SSC scenario [69].

- The declining phase of the exceptional TeV flare in PKS 2155−304 exhibited a cubic relation between the VHE and X-ray flux variations, which was even more challenging for one-zone scenarios and showed an inevitable presence of two or more electron populations [24].

- Finally, the recent X-ray polarimetric observations of the nearby bright HBLs with Imaging X-ray Polarimetry Explorer (IXPE; [70–72]) clearly showed a requirement of the inhomogeneous and/or multizone emission region with shock fronts, turbulence and magnetic reconnection (see Sections 3.2–3.4 for the corresponding discussions).

Inhomogeneous SSC variability models of increasing sophistication progressively overcome the problems of one-zone scenarios. For example, an instantaneous particle injection was replaced by variations in the injection rate by [57], which propagate throughout the emission zone and produce variations in the MWL flux, and the light-travel effects on the radiation from the different area were also taken into account. The authors of [73] constructed so-called one-zone expanding leptonic model where the emission region moves through the the jet and undergo a gradual expansion. Consequently, (ultra)relativistic electrons are the subject of the adiabatic, synchrotron and IC losses. A similar scenario was adopted by [74] to model the MWL flaring behavior of Mrk 421 during December 2016–June 2017.

*2.2. Multi-Zone SSC Scenarios*

*Inhomogeneous* and *multi-zone* SSC models provide a more realistic representation of the jet zone, which can be significantly extended. However, this is frequently achieved at the cost of a less detailed characterisation of the particle energy distribution [5]. The requirement of such models emerge when the one-zone SSC scenario fails to model the observed SED satisfactorily owing to the reasons as follows: (i) the sources of the MWL emission can be distributed along the jet; (ii) the emission zone propagates along the jet axis; (iii) there might be an electron populations characterized by different acceleration and cooling timescales.

Ref. [75] presented a model based on the SSC scenario by taking into account the time delays with which one observes the variability. Therefore, the source was split into smaller

one-zone models which evolve autonomously. Namely, (1) for each zone, the IC upscatter is based only on the synchrotron photons produced locally; (2) the electron energy losses are related only to these photons. This model was satisfactorily adopted for the MWL SED of Mrk 421 constructed from the MWL campaign of 1994 May, but it does not produce the EIC emission based on those synchrotron photons which come from other parts of the source at retarded times and, therefore, can not be valid for any $\gamma$-ray flare.

A significant improvement was made by [76], who calculated the SSC emission from a particular area of an inhomogeneous source as follows: one accounts for the synchrotron emission produced in the different source area and reaches the given location at retarded times. This produces more realistic SSC light curves and broadband SED, although in the case when the SSC losses (assumed to be a local process) are negligible. The basic challenge for this inhomogeneous multizone model is neglecting SSC losses due to the time-delayed photons coming from other parts of the emission zone. A further advance was made by [57], which presented such an inhomogeneous model that incorporates the effects of the non-local, time-delayed emission on the SSC losses at the given location. They assumed the presence of a relativistic EED injected in a "pipe" where electrons flow downstream and undergo a radiative cooling. One introduces variations in the injected EED which propagate downstream and are reflected in a frequency-dependent variability. Within this model, the orphan $\gamma$-ray flares can be obtained by assuming an increase in the injection of lower-energy electrons (in contrast to the EED's VHE tail).

As noted above, 1ES1959+650 underwent an orphan TeV flare in 2002 June, and [77] explained this event within the SSC scenario incorporating an inhomogenous emission zone. Namely, the primary flare (emerging in both X-ray and TeV $\gamma$-ray bands) is due to the injection of nonthermal electrons/positrons. However, when the jet is not uniform and contains different "patchy" area, X-rays produced within the primary flare undergo a scattering at the jet's dense region. This will result in strong increase in the TeV flux by IC upscatter, which can be observed as an orphan TeV flare since there should be a delay with respect to the primary flare.

During the giant 2006 flares of PKS 2155–304, the $\gamma$-ray spectra from the non-flaring nights were modeled within one-zone SSC model incorporating only small changes in the physical parameters by [78]. However, the VHE spectral and flux evolution during the flaring time windows were modelled by adopting a multi-zone SSC scenario. The latter succeeded in the interpretation of the hourly VHE variability, and a clear connection between the high activity in $\gamma$-rays and long-term increase in the lower-energy bands was deduced. Moreover, Ref. [24] explained the cubic relation between the TeV and X-ray variabilities exhibited by the source during the declining phase of this source as follows: probably, there was the appearance of a new $\gamma$-ray flaring component in this phase, which was strongly Compton-dominated and was, therefore, emitting few synchrotron emissions ($L_{IC}/L_{syn} \sim 10$). This component could be also very compact (of the order of several Schwarzschild radii), or dominated by external IC-upscatter of the photon coming from other jet regions.

Ref. [65] obtained a satisfactory fit of the broadband SED of Mrk 421 when introducing additional emission zones for the uncorrelated X-ray and TeV-band flaring activity in 2004. For a similar behavior observed during the TeV outburst in 2010 February, Ref. [79] adopted a two-zone scenario, where the larger zone was responsible for the stationary emission. The smaller emission zone was at the edge of this area, characterized by a transient turbulence and producing the variable emission.

Ref. [63] adopted a two-zone SSC model with different electron densities (outer and inner regions) for six different time windows from the MWL observations of 1ES 1959+650 in 2016. One assumed the presence of stronger magnetic field in the outer blob, amplified by the passing shock. The second, inner blob was characterized by a narrow EED and spectral hardening during the flaring periods, possibly owing to a stochastic acceleration process via Fermi-II process (see Section 2.3).

Mrk 421 underwent a TeV band flare a factor of ten on timescales of several hours in 2017 February, while only a moderate enhancement in the X-rays was observed [80]. The broadband SED from this event was modeled in the framework of a two-zone leptonic scenario, according to which the TeV-flare was explained by introducing a compact second blob, which contained ultrarelativistic electrons characterized by a relatively narrow range of Lorentz factors $2 \times 10^4$–$6 \times 10^5$. This EED was suggested to result from stochastic acceleration in a turbulent jet medium, yielding a quasi-Maxwellian energy distribution (see Section 2.3 for details).

While two-component models frequently show a better fit with the observed SEDs of HBLs, it is problematic to constrain the free parameters due to their large number. Ref. [80] derived such constraints from the VLBI and radio polarization observations of four HBLs (PKS 1424+240, 1ES 1727+502, 1ES 1959+650, 1ES 2344+514) and selected seven epochs from the period 2013–2016 for these objects based on the TeV variability (e.g., low, intermediate and high TeV-states for 1ES 1959+650). The corresponding broadband SEDs were modelled within the two-zone SSC scenario, where the two co-spatial emission zones are situated at the VLBI core (separated by several parsecs from the central SMBH) and the constraints on jet physical factors (magnetic field strengths, Doppler factors etc.) derived from the VLBI observations were used for this purpose.

As noted above, the presence of a higher-energy SED peak above 1 TeV poses a challenge to the one-zone leptonic model. Nevertheless, VHE $\gamma$-ray spectra should steepen in the process of the electron acceleration to the ultrarelativistic energies, accompanied by declining in the energy densities of synchrotron seed photons valid to be upscattered in the Thomson regime and increasing dominance of the IC scattering in the KN regime [15]. Ref. [81] proposed a two-component model to explain the extreme high-energy peak of UHBLs: the internal component is produced by the SSC mechanism, while the external one is related to the interaction between the relativistic protons (accelerated within the jet) and the photons from the cosmic microwave background (CMB). Within the latter process, electron-positron pairs will be produced, which upscatter soft photons to $\gamma$-ray energies.

Ref. [82] presented a model where particles are accelerated at the recollimation shocks, triggered during the recollimation of the UHBL jet by the external plasma. While the EED generated at the single shock front was sufficient to reproduce the SED of relatively less extreme EHBL sources, the same was not possible for the hardest sources (e.g., in the case of 1ES 0229+200) and the existence of multiple recollimation shocks was proposed. In fact, the latest simulations of the recollimation process in weakly magnetized jets showed that the jet flow is a subject of a rapidly growing instability after the first recollimation shock. Consequently, it becomes highly turbulent and decelerates, preventing the formation of multiple shocks [83]. Based on these findings, Ref. [84] proposed a revised scenario for UHBLs: electrons are accelerated at the recollimation shock font via the Fermi-I mechanism and, subsequently, gain energy through the stochastic acceleration in the turbulent downstream medium. As a result, the observer will record the emission from the entire downstream area, where electrons are at different stages of acceleration. This scenario was applied to 1ES 0229+200 the broadband SED of which was satisfactorily described by using the reasonable values of the jet physical parameters.

Ref. [85] reported the presence of a narrow VHE spectral feature at $\sim$3 TeV for Mrk 501 obtained on 19 July 2014, detected with a significance of higher than $3\sigma$. In order to explain the origin of this feature, a structured jet model was proposed: there could be two jet regions producing the $\gamma$-ray emission by means of the SSC mechanism. One region was characterized by an extremely narrow EED, and the second, smaller-size emitting region was additional to the first (larger) one. Two different geometries were considered: (1) these regions were co-spatial, with the second blob is embedded within the first region; (2) the regions were not co-spatial. In the first case, the photon density within the smaller blob should be sufficiently high, while the external photon field produced by the larger region was negligible for the IC-scattering and for the electron-positron pair creation. Otherwise, the interaction of the relativistic electrons and the emitted gamma rays from the small

blob with the synchrotron emission from the large region would broaden and absorb the spectral TeV feature. Within the second scenario, the smaller region should be situated closer to the observer (than the larger one) to avoid the $\gamma$-ray absorption by the intense infrared photon field. Even in this situation, a large Doppler factor is required for this blob to produce the aforementioned spectral feature due to the very narrow EED (in contrast to the second region).

A variety of multi-zone lepton scenarios is the so-called spine–sheath model, where the jet contains a quickly moving spine (dominated by the electron-positron plasma) within a less relativistic sheath (possibly, with a significant baryonic content; Ref. [86] and references therein). This scenario is primarily devoted to explain the high bulk Lorentz factors obtained from the SED modelling of HBLs, while the significantly lower values were measured via the radio interferometry. The spine-sheath model assumes a fast variability from a thin spine against slowly variable emission from the sheath. The latter provides a low-energy photon field for the external IC upscatter to the $\gamma$-ray energies inside the spine by the local ultrarelativistic leptons.

### 2.3. Hadronic and Leptohadronic Processes

Generally, leptonic models provide a relatively economical approach with respect to the free parameters and the jet energy requirements [5]. However, hadronic models are of particular interest whenever leptonic scenarios face difficulties, and there are various reasons to introduce the hadronic scenarios which are capable of contributing to the observed HBL SEDs. For example, one of the open problems is the origin of ultra high-energy cosmic rays (UHECRs) and high-energy neutrinos. HBLs represent one of the potential emitters of UHCRs; their low-power jets can provide the suitable acceleration [87]. The inclusion of hadronic components in the HBL emission origin is particularly important since it allows to estimate the possible contribution to the flux of neutrinos and UHECRs. (Lepto)hadronic models are capable for discerning potential sites of the UHECR and connect them with the expected emission of the VHE neutrinos and photons [5]. Moreover, there are some evidences from observations and modelling that relativistic blazar jets should contain a significant hadronic component (see, e.g., [88]).

In the framework of the hadronic models, the lower-energy SED component is an electron-synchrotron emission, while the relativistic hadron population contributes to the $\gamma$-ray emission of HBLs [8,20]. Both the electron and proton populations are accelerated to ultrarelativistic energies (e.g., at relativistic shock fronts; see Section 3.2), until protons exceed the $p\gamma$ (photo-pion) production threshold on the soft photon field existing in the emission zone [7,82]. There are different possible scenarios for producing higher-energy emission as follows:

- Proton-synchrotron. In the framework of the so-called *synchrotron-proton blazar* (SPB) model ([89,90] and references therein; [7]), a significant portion of the jet kinetic or magnetic power is used to accelerate protons in a strongly magnetized environment to the aforementioned threshold and various synchrotron-emitting pair cascades may develop [8,20]. For this purpose, the acceleration of protons to the energies ($E_p^{\max} \gtrsim 10^{19}$ eV and Lorentz factors $\sim 10^{10}$) is necessary for obtaining a dominant proton-synchrotron emission in the $\gamma$-ray energy range. In turn, this requires high magnetic fields of $\sim 1$–100 G in order to constrain the Larmor radius smaller than the size of the emission region itself [5,7]. Alternatively, significant hadronic emission can be produced within weaker magnetic fields combined with large particle and/or photon densities [16]. In such a situation, the energy density of relativistic protons needs to largely exceed that of relativistic leptons to contribute significantly to the $\gamma$-ray domain. This can be achieved by imposing the specific requirements on the acceleration process [5]. In the case of the aforementioned magnetic field values and Doppler factor $\delta = 10$–50 in the HBL jets, the proton–synchrotron peak frequency is expected in the range of 10–100 GeV [7]. The number of free parameters of the

proton–synchrotron scenario is significantly larger than the SSC one (amounting to 14; see, e.g., [7]).

- Modified proton-synchrotron. In the later versions of SPB model, the synchrotron radiation of secondary muons and mesons was also taken into account [91] and references therein; [20]). First of all, one expects a photo-pion production process ($p\pi$) where a photohadronic $p + \gamma$ interaction yields either $\pi^0$ or $\pi^\pm$ mesons. For this purpose, the photon energy in the proton frame should be higher than about 145 MeV [7]. Gamma-ray photons can be obtained from the $\pi^0$-decay process ("$\pi^0$-cascade"), or produced by electrons from the $\pi^\pm \to \mu^\pm \to e^\pm$ decay ("$\pi^\pm$-cascade"). One expects also the proton-synchrotron emission ("$p$-synchrotron cascade"), as well as the $\mu$-, $\pi$-and $K$-synchrotron photons ("$\mu^\pm$-synchrotron cascad"; [7]). Refs. [16,91] demonstrated that the $\pi^0$ and $\pi^\pm$ cascades initiated by ultra-high energy protons generate featureless $\gamma$-ray spectra, in contrast to $p$-synchrotron and $\mu^\pm$-synchrotron processes: the latter produce a two-component $\gamma$-ray spectrum, i.e., the muon synchrotron radiation emerges as a third SED component, at higher energies than the synchrotron radiation by the parent protons [7]. Generally, direct proton and $\mu^\pm$ synchrotron radiations are thought to be the main contributors to the higher-energy SED "hump", while the low-energy component is synchrotron radiation from the primary electrons, along with some contribution from the secondary electrons generated by the aforementioned cascades [8,20]. Electrons and positrons produced in the decay of charged pions have extremely high-energy, and their synchrotron radiation can reach even PeV energies [7]. Generally, the jet emission region is "opaque" for first generations of secondary particles and $\gamma$-rays, leading to successive reiterations of the above-described cascades [16]. The decay of neutral pions can produce ultra-high-energy (UHE, $E > 100$ TeV) $\gamma$-rays or so-called PeV-photons [92]. However, these photons do not reach us, being absorbed via pair-production both in the jet, or during the propagation in the (inter)galactic medium [7]. Photo-meson production is characterized by a key property: neutrinos are produced along with photons, escaping the emission zone without any absorption or energy losses and their detection directly indicates the presence of highly-relativistic protons in the jet, as well as is capable for constraining the model key parameters [7]. The proton-proton interactions are thought to be negligible in the SPB models, since this mechanism requires very high particle density and the extreme jet powers for producing a significant $\gamma$-ray emission [5].

- Bethe–Heitler pair production. A photohadronic interaction between relativistic protons and photons may also result in the Bethe–Heitler pair production as $p + \gamma \to e^\pm$ ([93] and references therein). This process is in competition with the photo-meson production, although it needs significantly lower lower energies: the threshold for the Bethe–Heitler pair production is lower than the photo-meson one by a factor 0.004 [7]. Consequently, the generated pair produces a lower-energy emission compared to the photo-meson cascades. Namely, the simulations of [94] showed the appearance another higher-energy SED component due to this pair production in the energy range 40 keV–40 MeV (so-called three-hump SED). Although the corresponding peak luminosity can not be always comparable to that emitted above 40 MeV, this keV–MeV SED component may still be observable (not being hidden from other components). Therefore, observation of the three-hump hump SED may indicate a viability of the leptohadronic scenarios.

Another key characteristic of photo-meson interactions is the creation of neutrons, which escape the emitting region without interacting with magnetic fields and Bethe–Heitler pair production [7]. These neutrons can transfer a significant amount of energy at large distances downstream from the jet and decay into protons, radiating synchrotron photons in the presence of magnetic fields. Consequently, the existence of two separate, causally connected hadonic emission zones with a significant separation is possible [7,95].

The spectra of HBLs are relatively well reproduced by proton-synchrotron-dominated SPB models where the intrinsic primary synchrotron photon energy density is small,

consistent with the low bolometric luminosity of those objects [5]. As the synchrotron photon energy density increases, relativistic protons undergo increasing energy losses from the $p\gamma$ pion production process. Consequently, the contributions from the $\pi^{\pm}$ and $\mu^{\pm}$ cascades become progressively dominant at higher energies. On the other hand, this process yields a decrease in the peak energy of the $\gamma$-ray component [96].

Generally, the hadronic processes are relatively inefficient in point of the produced energy (compared to the leptonic scenarios) and, moreover, require the extreme, super-Eddington jet powers $P_{\text{jet}} \sim 100 L_{\text{Edd}}$ (so-called energy crisis), where $L_{Edd}$ is the Eddington luminosity [5,97,98]. Consequently, the hybrid *lepto-hadronic* models provide a more reasonable physical approach [8,16,20,82]. Moreover, this energy-crisis hadronic scenario is more inherent to flat-spectrum radio-quasars (FSRQs), while it is less prominent for low-luminosity HBLs and hadronic solutions with $L_{\text{jet}} < L_{\text{Edd}}$ can be achieved [7]. Nevertheless, the detection of the muon neutrino with the most probable energy of $\sim$290 TeV from the IBL source TXS 0506+056 (or, more plausibly, a FSRQ object; see [99,100]) revived and deepened interest in these models. Note that [101] modeled the broadband SED of TXS 0506+056 from the neutrino detection epoch by means of a leptohadronic scenario, in which the Bethe–Heitler and pion-decay processes produce the X-rays and VHE $\gamma$-ray emissions. The observed neutrino flux was used for the model constraining. Ref. [102] investigated a connection between HBLs, Ice-Cube neutrinos, and UHECRs and found a probability $\sim$0.18% ($2.9\sigma$) after compensation for all the considered trials. Moreover, they deduced that HBLs can account only for $\sim$10% of the UHECR detections.

Ref. [103] explained a lack of the $\gamma$-ray activity along with the X-ray ones in some HBLs by production of the TeV–PeV neutrinos in the case that the X-ray flares are powered by the proton-synchrotron mechanism: neutrinos are expected from the photo-meson interactions of ultrarelativistic protons with their own synchrotron radiation, while the MeV-to-GeV $\gamma$-rays emission is produced within the synchrotron-dominated electromagnetic cascades.

Ref. [104] modeled the SEDs of those HBLs thought to be counterparts of the IceCube-detected neutrinos (Mrk 421, PG 1553+113, 1ES 1011+496, H 2359-309, 1RXS J054357.3-553206 and 1H 1914-194), adopting an one-zone leptohadronic model. It was concluded that the model fits with these SEDs by using the reasonable values of the jet physical parameters (e.g., $B = 0.05$–$5$ G, $\delta = 18$–$31$, $\gamma_{\text{e,max}} = 8 \times 10^4$–$2 \times 10^6$). In the case of Mrk 421 and 1H 1914$-$194, a good agreement between the model-predicted and the detected neutrino fluxes ( from the events with IDs 9 and 22, respectively) was found. Note also that [105] reported the *AGILE* detection of a candidate $\gamma$-ray precursor to the ICECUBE-160731 neutrino event, which was identified with the X-ray source 1RXS J141658.0$-$001449. Based on the X-ray-to-radio flux ratio, the object was concluded to have properties typical to the HBL sources. However, no further identification of 1RXS J141658.0$-$001449 was performed (e.g., detection of the featureless optical spectrum). The aforementioned model was adopted for the multi-epoch modeling of TXS 0506+056, including the time windows corresponding to the neutrino detection instances from this object [106]. For the same purpose, [107] developed the `SOPRANO` code which included all hadronic processes yielding high-energy neutrinos. This code (along with the `LeHa` code developed by [16]) was adopted by [51] to model the broadband SEDs of Mrk 501 corresponding to the different time windows of the period 2017–2020 and evaluate the expected neutrino flux from this object.

In the leptohadronic scenarios, the proton–synchrotron component is relatively suppressed via imposing the magnetic field to be not higher than 1 Gauss and the SSC emission dominates in the $\gamma$-ray output (in the case of HBLs). The simplest case is the one-zone leptohadronic model: all radiation mechanisms are operating in the same emission zone and external photon fields are negligible [7]. Here, relativistic protons produce secondary leptons via the p-$\gamma$ interactions over the electron synchrotron photon field. The emission from such leptons can contribute to hard X-rays (as Bethe–Heitler component) and in the TeV band (as photo-meson component; see [104]). Such a model was presented by [108] along with a new extended hadroleptonic code `ExHaLe-jet`, which considers simultaneously the processes related to relativistic protons and electrons. Within a predefined geometry and

bulk flow physical parameters, the particle evolution was simulated. Highly relativistic secondary electrons (and positrons) are created through the $\gamma\gamma$ and Bethe–Heitler pair productions, as well as during the pion/muon decay. The ratio of protons to these secondaries was assumed to decrease with distance from the jet base. For particle–photon interactions, all internal and many external photon fields were considered. Note that the external fields were concluded be the more important source for particle–photon interactions leading to the neutrino production. Note that this result is related to the fact that lepto-hadronic solutions also face energetic issues, especially when one tries to maximize the neutrino output and, consequently, the required jet power can quickly become very high [7].

Ref. [109] adopted an one-zone leptohadronic model for the X-ray and $\gamma$-ray flares shown by Mrk 421 in 2001 March. First, they performed a preflare SED modeling, using the different leptohadronic scenarios. Afterwards, by introducing small-amplitude variations in the injection rate and in the maximum particle energy, the flaring state SEDs were reproduced. Note that the models incorporating the pion-decay processes successfully reproduced the observed quadratic relation between X-ray and TeV variabilities. Ref. [51] adopted a leptohadronic model for Mrk 501, in which the high-energySED component represents a combination of both leptonic (IC) and hadronic (emission by cascades triggered by hadronic interactions) processes. It was assumed that the bulk of the high-energy SED component is generated by the SSC mechanism, while the hadronic output are subdominant and can emerge (and even dominate the SED) in hard-X-rays, filling in the SED dip, and in the VHE band. In the presented framework, the proton-synchrotron emission is very suppressed owing to the lower magnetization of the emitting region compared to that required otherwise. A similar combination was adopted by [110] to reproduce the broadband SED of the HBL source Mrk 180.

Ref. [111] proposed so-called *hadronic synchrotron mirror* model for explaining the orphan TeV flare of 1ES 1959+65 in 2002 June: X-rays produced in the process of the primary $\gamma$-ray flare were reflected by a plasma cloud (situated nearly in the direction of the jet propagation), then collided with the jet protons and, consequently, the pion-poduction cascade was developed which yielded the observed orphan TeV flare. However, this model requires very high proton density in the jet and hadronic jet power. Nevertheless, the model can be physically reasonable, if one takes into account the effects related to the emission zone approaching to the mirror [77]. This scenario was adopted by [53] for the MWL observations of 1ES 1959+65 in 2012 April–June. A leptohadronic model was adopted by [54] for the highest TeV states of 1ES 1959+650 recorded in 13–14 June 2016, although requiring extreme magnetic field ($B \sim 100$ G) and very high values of the jet power ($\sim 10^{46}$ erg s$^{-1}$).

As noted above, one-zone SSC models are problematic for UHBLs, since they require large Doppler factors as well as extremely high minimum Lorentz factors for the EED. However, the UHBL SEDs were modeled by [16] within a leptohadronic framework, without adopting the extreme Doppler factors the aforementioned extreme minimum Lorentz factors (adopting $\delta = 30$ and $\gamma \sim 10^{2-3}$). In the case of the significant proton-synchrotron radiation, magnetic fields $B \sim 1$–100 G and maximum proton energies $E_p^{\max} \lesssim 10^{19}$ eV were derived. In the case of the synchrotron emission from the $p\gamma$-induced cascades, the range $B \sim 0.1$–1 G was required. Moreover, the deduced jet powers were mostly sub-Eddington, in contrast to previous hadronic modelings (see the corresponding discussion above). A caveat of the [16] model is the very hard spectra of injected particles, required for the co-acceleration of leptons and protons. The hard TeV of 1ES 0229+200 was explained by [112] in the framework of one-zone hadronic model where $\gamma$-ray emission is produced via the neutral pion (from proton-proton interaction) decay, but at the cost of adopting a small radius of the radiation zone than the Schwarzschild radius of the central SMBH.

When both leptonic and hadronic radiative models provide similarly good fits to the observed broadband SED, they can be distinguished from the HBL's temporal behavior. While the one-zone SSC model is characterised by correlated variations in both the synchrotron and higher energy ranges, time-dependent hadronic models require the solution of

a system of coupled differential equations of the kind $\partial N_X(t, E)/\partial t = Q_X(t, E) - L_X(t, E)$ for each particle species X (protons, photons, neutrinos, leptons), with $Q_X(t, E)$ and $L_X(t, E)$ to be the injection and loss terms, respectively [7]. Due to the complexity, the time-dependent hadronic modeling have been used relatively rarely (e.g., [113–116]). For example, Ref. [117] modeled the broadband SED and MWL lightcurves of the HBL source 1ES 1011+496 using a hybrid leptohadronic model, taking all relevant processes into account (acceleration and synchrotron emission of both electrons and protons, IC scattering, photo-hadronic interactions and $\gamma\gamma$-pair production). This model yielded a more satisfactory representation of the target's VHE flare compared to the pure leptonic modeling. Although the two SED components are produced by two distinct particle populations within the proton–synchrotron models, the observed correlation between the synchrotron and $\gamma$-ray variabilities can be achieved by assuming that electrons and protons are energized by the same acceleration mechanism (as done by [109] for Mrk 421; see above). Otherwise, hadronic and hybrid models as useful to reproduce the absent MWL correlation, which is a challenge for the leptonic one-zone scenarios (see the discussion related to the "orphan" flares). For example, ref. [118] proposed that the low energy tail of the SSC photons (1–8 MeV) of 1ES 1959+650 served as the target for the Fermi-accelerated high energy protons of energy $\lesssim$100 TeV, producing the TeV photons through the decay of neutral pions from the $\Delta$-resonance during the orphan TeV flare in 2002 June. Later, this model was adopted for modeling the GeV–TeV flaring episodes of 1ES 0229+200, 1ES 0347−121, 1ES 0806+524, Mrk 501 and HESS 1943+213 [119,120]. This model was expanded into the two-zone photohadronic interpretation, adopting different emission zones below and above the threshold energy of 1 TeV for 1ES 1959+650 (for the time window 2016 November 19–21; [121]), Mrk 501 (2005 May–July and 2012 June; [122]), Mrk 421 (2010 March; [123]) and 1ES 2344+514 (several $\gamma$-ray episodes; [124]).

On the other hand, it is generally problematic for the leptohadronic models to deal with a very rapid TeV-band variability of HBLs: the radiative cooling time scales of protons is of the order of several days even in the case of the magnetic fields of $\sim$10 Gauss and typical Doppler factors $\delta = 10$ (adopted for HBLs; see, e.g., [7]). However, the rapid $\gamma$-ray variabilities, observed on timescales shorter than the proton cooling time, can be attributed to the geometrical effects (see, e.g., [16,90] and Section 3.3 for the jet-in-jet scenario).

*2.4. Magnetospheric Vacuum Gaps, Curvature Emission and EIC Scattering*

SMBHs are widely accepted to be the central engines of AGNs (including HBLs), where particles should be accelerated by extracting their rotational energy via the Blandford-Znajek (BZ) process [3]. Moreover, the SMBH magnetospheres could be the sites for the origin of strong and fast VHE flares, depending on the importance of the $\gamma\gamma$ absorption. Note that this effect can be weaker in HBLs which are widely accepted to possess sub-luminous accretion disks operating in the RIAF regime and sufficiently low magnetic fields (see above).

The BZ-mechanism operates in the force-free magnetosphere containing the high-energy plasma. The latter is continuously removed from there during the jet collimation process and one expects the appearance of an area with a charge deficit, so-called vacuum gaps (or spark gaps). In these gaps, the charged particles can be accelerated up to ultrarelativistic energies along the open magnetic field lines[125]. Vacuum gaps can appear at the jet base, at a few gravitational radii from the horizon and produce the electron-positron plasmoids which are capable of triggering a fastly variable VHE $\gamma$-ray emission [5]. Moreover, charged particles can be generated in these gaps via the pair-production cascades [125]. Consequently, vacuum gaps allow us to draw conclusion about the physical conditions in the vicinity of the SMBH horizon.

The TeV-band variability could be enhanced by sharp changes in the physical conditions (e.g., the local accretion rate, abrupt changes in the disc emission, magnetospheric currents) throughout the magnetosphere [126]. Curvature emission is thought to be one of the possible mechanisms producing a fast TeV-band flare [127]. Namely, when electron-

positron pairs are created by means of the $\gamma\gamma$-interactions, these particles will be accelerated towards the opposite directions along the field lines (electrons and positrons are accelerated outwards and inwards, respectively) and produce $\gamma$-rays via curvature emission, as well as by external IC upscatter of soft photons coming from the inner accretion disc [127].

Note that photons emitted by the innermost disc parts can enter the SMBH magnetosphere and undergo the $\gamma\gamma$ annihilation and supply the vacuum gap with these particles [128,129]. Electron-positron pairs will be accelerated fastly in the gap owing to the large potential drop produced by the rotating SMBH, until reaching those values of the Lorentz factor for which the energy gain is balanced by curvature radiation or EIC losses [129]. The amount of the gap-born TeV-band emission (and, hence, the $\gamma$-ray luminosity of the vacuum gap) depends on the gap size along the magnetic field lines, and increases with the declining accretion rate. It was found that the gap width is not smaller than $0.01r_s$ in the case that the density of "seed" charges is below the Goldreich–Julian (GJ) value [127]. When the accretion rate becomes $\dot{m} < 10^{-4}$ ($\dot{m} = \dot{M}/\dot{M}_{\text{Edd}}$, with $\dot{M}$, the accretion rate; $\dot{M}_{\text{Edd}}$, the Eddington accretion rate), the SMBH magnetosphere becomes charge-starved and a vacuum gap is switched on [130]. In the case of the extremely rotating SMBH, the gap is though to produce a large VHE $\gamma$-ray emission and the TeV-band spectrum can extends to higher frequencies with the increasing flux [129]. In the local frame, the peak energy of the curvature spectrum should be limited to $\sim 50(\dot{m})^{3/8}$ TeV, which is below 1 TeV for $\dot{m} \lesssim 10^{-4}$ [129].

However, the TeV luminosity of vacuum gaps is limited by the pair-production process during the interaction of TeV photons with the ambient radiation field. If this occurs outside the gap, the created pairs move away from the gap as the secondary pairs and emit secondary photons via the IC and the synchrotron processes [130]. In the case that these secondary photons materialize within the magnetosphere, the so-called tertiary pairs are produced and, in turn, emit the tertiary photons via IC and synchrotron processes, and so on. The multiplicity of this process was found to depend on the accretion rate [129]. Ref. [130] suggested that such a cascade can propagate up to $60r_g$. First of all, this process is initiated by those TeV photons which have energies higher than 10 TeV (up to $\sim 10^3$ TeV), for which the $\gamma\gamma$-optical depth is much larger than for the lower-energy photons: the pair-production opacity drops with decreasing $\gamma$-ray energy and becomes sufficiently small below 10 TeV that allows the photons at this energy range to escape the gap and, eventually, the magnetosphere [129].

The observed $\gamma$-ray spectrum depends on the spectrum of soft (scattered) photons and the pair cascade process. Moreover, an increase in the curvature radius of the gap magnetic field lines will boost the maximum energy of accelerated electron-positron pairs. In turn, this will lead to the broadening of the TeV emission spectrum [129]. As noted above, the spark process in the gap can be highly intermittent: the TeV-band luminosity and the variability amplitude depend on the pair-creation opacity (which, in turn, is sensitive to the soft radiation produced by the innermost disc regions). Within this mechanism, a fast TeV variability is expected even in the case of the moderate changes in accretion rate: this will trigger nonlinear fluctuations of the gap potential, induce intermittencies in the pair-production opacity and the strength/geometry of the magnetic field advected by the accretion flow, changing also the co-aligned electric field nonlinearly [129]. The simulations showed that such changes can produce the delayed TeV flares, mainly contributed by the curvature emission. The flare rise duration should be of the order of the light-crossing time of the gap, although the exact shape of the TeV-band light curve depends on the gap's separation from the event horizon: one expects that a strong lensing will significantly affect the observed light curve [127].

As noted above, two mechanisms are thought to be responsible for the $\gamma$-ray emission by the electron-positron pairs accelerated in the gap: the EIC scattering and the curvature radiation [127]. The latter represents a synchrotron variant for charged particle moving along curved magnetic field line: the produced radiation is related to the field line curvature, not to the gyro-acceleration. However, there are some distinctions between these

mechanisms: (1) for the curvature emission from a single (ultra)relativistic particle, the total emitted power and characteristic frequency $P_{\text{curv}} \sim \gamma^4$ and $\nu^c_{\text{curv}} \sim \gamma^3$ (versus $P_{\text{syn}} \sim \gamma^2$ and $\nu^c_{\text{syn}} \sim \gamma^2$ for the synchrotron emission) [127].

An alternative hypothesis to explain the narrow spectral feature at 3 TeV in Mrk 501 (reported by [54]) is based on the gap emission from the electrons accelerated to energies of about 3 TeV in a sporadically active magnetospheric vacuum gap close to the central SMBH [54]: there could be electromagnetic cascades triggered by the interaction of relativistic electrons/positrons with emission line photons coming from the photoionized gas clouds. Even though Mrk 501 is an HBL source and should not have a significant BLR, [54] speculated the possibility that gas clouds from the inner parts of the host galaxy intruded into the AGN. Along with the EIC upscatter, there could be the cascades from the Breit-Wheeler (BWPP; [131]) pair-production process incorporating collisions between the local high-energy and low-energy ambient photons and creating electrons which are capable for the IC-scattering (see [54] for details). Consequently, the corresponding VHE emission manages to escape outward and a narrow TeV-band component was formed, superimposed on the SSC emission from the distinct (larger) jet emission zone.

As discussed in Section 2.2, the uncorrelated VHE flare of Mrk 421 observed in 2017 February was explained within two-zone SSC scenario: there was a compact second blob containing highly energetic electrons which were characterized by a narrow range of Lorentz factors. In turn, this population was suggested to result from an electromagnetic cascade initiated by electrons accelerated in the magnetospheric vacuum gap of Mrk 421 [74].

## 3. Variability Mechanisms

Because of the very small angular size, it is not possible to directly resolve the emission zone. Therefore, information about the spatial structure of this jet region can be obtained through the MWL variability studies. Particularly important is the variable emission produced by the EED's highest-energy part (X-rays and $\gamma$-rays) since these electrons cool very quickly and can exist only close to the site they were produced.

The intense MWL observations of HBLs revealed a complex structure in the $\gamma$-ray variability. For example, Mrk 421 underwent two dramatic outbursts of the TeV emission in 1996 May: the first flare showed a flux-doubling time of $\sim$1 h and, eventually, the TeV flux increased by more than a factor of 50, making Mrk 421 the brightest TeV source in the sky. During the second outburst, the brightness boosted by a factor of 20–25 in about 30 min [132].

On some occasions, the TeV variability of HBLs was extremely rapid. For example, Mrk 501 showed flux-doubling time as short as 2 min during the strong flaring activity recorded in 2005 May–July, along with the longer-term VHE variability by an order of magnitude during the entire campaign [18]. A similar range was observed for PKS 2155-304 in 2006 July, when the well-resolved flares on timescales of $\sim$200 s were detected (see [17] and Figure 2). These instances implied highly relativistic sub-parsec scale flows ($\delta \sim$ 50–100; [133]), and the emitting region (constrained by means of the causality relation $R < ct_{\text{var}}\delta/(1+z)$; see, e.g., [54]), was comparable or even smaller than the size of the central SMBH horizon (even for high jet bulk Lorentz factors). These observation signatures demonstrate that the HBL jets can be structured on very small spatial scales that are unresolved by the current $\gamma$-ray instruments.

The power spectral density (PSD) represents an important tool for characterizing the nature of flux variability: it provides a measure for the contribution of different timescales to the variability power by quantifying the amount of variability power as a function of temporal frequency ($\nu \sim 1/t$; [41,134]). Similar to other AGN subclasses, one of the major observational characteristics of $\gamma$-ray variability in HBLs is their power-law type behavior. Namely, $PSD(\nu) \sim \nu^{-\beta}$, with $\beta \geqslant 0$ associated with a random-walk behavior in the time domain (i.e., nonperiodical variability; [134]) as follows:

- white noise ($\beta = 0$);

- pink or flicker noise ($\beta = 1$);
- red or Brownian noise ($\beta = 2$).

In the case that the source is showing a broken power-law PSD with the break frequency $\nu_b$, then the flux variability is characterized by the characteristic timescale $t_{char} \sim 1/\nu_b$ which also can be a variability period ([41,135]; see also Section 3.1). For example, PKS 2155-304 exhibited a red-noise behavior with $\beta \sim 2$ during the aforementioned exceptional VHE flare [136]. The long-term VHE and HE observations of the source (the H.E.S.S. and *Fermi*-LAT data, respectively) demonstrated a a flicker noise with $\beta \sim 1$ [134]. Mrk 421 showed a pink-noise VHE behavior during the intense MAGIC observations during 2009 January–June [41], etc.

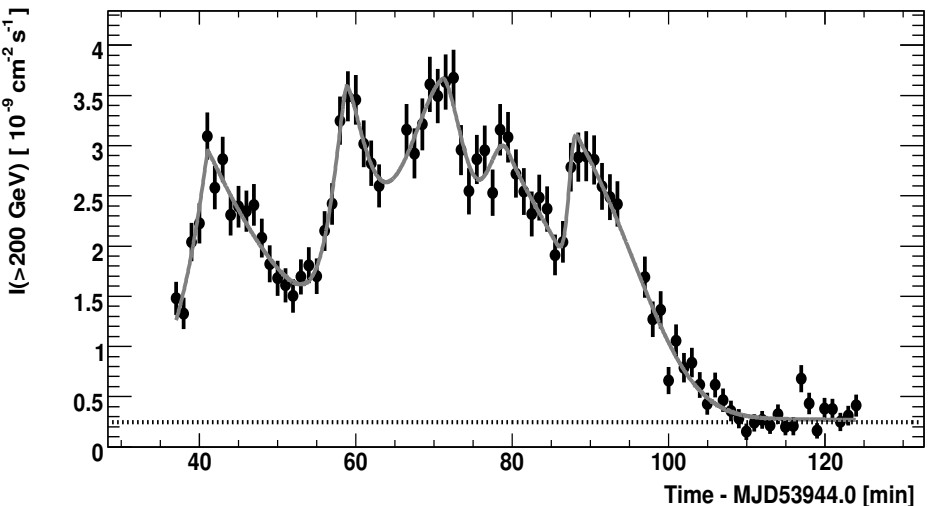

**Figure 2.** The VHE variability of PKS 2155−304 shown at the energies higher than 200 GeV shown on 28 July 2006. The horizontal line represents the Crab emission in the same energy range. Reproduced by permission of AAS from [17].

### 3.1. Variability Models and Quasiperiodic Flux Changes in HBLs

To date, several models have been proposed to explain the $\gamma$-ray variability of HBLs:

- *Shock-in-jet* scenario (e.g., [21,76]);
- *Jets-in-jet* model and *relativistic magnetic reconnection* [133,137,138];
- *Jet turbulence* [26,139,140];
- Instabilities in the magnetospheric gaps (see Section 2.4);
- Jet precession ([141–143]).

In the latter case, the system consists of a primary SMBH (with the associated accretion disk and jet nearly pointed to the observer), and a smaller-mass, secondary BH orbiting the primary one (see, e.g., [143]). In such a situation, a quasiperiodic flux variability may emerge due to the periodic change of the jet orientation towards the observer. Such a change can be caused by the two different effects [141]:

- The dominant effect (causing the jet angle to vary by the greatest amount) is simply an imprint of the SMBH orbital velocity on the jet: since the jet-carrying primary SMBH is moving along a circular orbit with the velocity V, the highly-relativistic ejected material is expected to have the same velocity component in the observer's rest frame. Consequently, the jet will precess with respect to the distant observer and the $\gamma$-ray emitting the region is observed at an angle $\theta_{obs}$ oscillating with an amplitude $\theta_{obs} = [2q/(1+q)][GM/Rc^2]^{1/2}$ and period $T = 2\pi(R^3/GM)^{1/2}$, with M—the system's total mass; q, the primary-to-secondary mass ratio; R, the separation between the components, assumed to be larger than a few Schwarzschild radii. Consequently, the jet's instantaneous shape will be helical, where radius of the coils increases linearly

with the distance from the primary SMBH and one should observe a quasiperiodic flux variability.

- The second-order effects can be caused by the general-relativistic deflection and Lense–Thirring precession. Namely, a Lense–Thirring precession ([144] and references therein) of the primary SMBHs acretion disc can be triggered by the gravitational field of the secondary SMBH. Consequently, the primary's jet is also expected to precess with the same period. However, the corresponding oscillation angle (and, hence, the amplitude of the periodic flux variability) will be significantly smaller than that caused by orbital movement of the primary SMBH, if these two SMBHs are separated by more than a few Schwarzschild radii. The general-relativistic effect causes a deflection of the relativistic ejecta's trajectory by the gravitational field of the secondary SMBH. Note that the general-relativistic effects are expected to be negligible on the few-years timescale [141].

Note that these variability mechanisms differ from the others in that they produce long-term (quasi)deterministic periodicity and trends, rather than stochastic variability (as discussed below).

Currently, the light curves compiled from the continuous *Fermi*-LAT observations form the most suitable database to search for the quasiperiodic variability in the HE $\gamma$-ray band. Among HBLs, PG 1553+113 is the primary candidate among HBLs for hosting a binary SMBH system. Initially, a HE periodicity with $\approx$2.2 yr was reported by [145] by using the Lomb-Scargle Periodogram (LSP) and continuous wavelet transform (CWT) for the *Fermi*-LAT observations of the source. However, the LSP peak is below the $3\sigma$ significance and the periodicity detection can not considered as highly-credible (see, e.g., [146,147]). A similar result was reported by [142] from 9-yr *Fermi*-LAT observations of the source, although no detection significance was evaluated. Ref. [146] adopted the PDS method and concluded that the constructed PDS for PG 1553+113 was compatible to noise, i.e., non-periodical variability was singled out with a significance level higher than 95%. Ref. [148] adopted different periodicity searching techniques (LSP, REDFIT, CWT etc.) for the LAT 2008–2017 observations of the source and deduced a period of 2.2 yr by the average significance higher than $4\sigma$. Moreover, the use of these traditional Fourier-like methods for the periodicity search (as done by [148]) was called into question by [149], which adopted the Gaussian process methods—CARMA and Celerite—for the same purpose. For PG 1553+113, possible evidence for the period of $\sim$800 days was found, with local significance of $\gtrsim$95%. However, the global significance was only 50–90% when constructing the LAT-band light curves with different time bins. A further improvement was done by [150] which adopted a Gaussian process modeling (along with the LSP technique) and various tests to conclude a quasiperiodic variability in the optical *R*- and LAT 0.1–200 GeV bands. The obtained $\sim$2.2 yr period was confirmed also by [151], which performed the analysis of the LAT data by adopting different methods for this purpose. Ref. [143] hypothesized that PG 1553+113 should posses a relativistic jet which rotates with a constant angular velocity around some axis, owing to the central binary SMBH (similar explanations were also presented by authors claiming the periodicity detection for the source). Consequently, the jet Doppler factor (and, hence, the observed brightness) undergoes a periodical variability. Based on this scenario, a light curve with 2.2-yr periodical oscillations was generated and compared to the 0.1–300 GeV light curve constructed from the 13.6-yr LAT observations of PG 1553+113. However, no attempt was made to evaluate the periodicity significance. Finally, Ref. [152] concluded that that periodicity detection significance becomes less than $1\sigma$ after the trial correction.

Ref. [148] reported the period of 1.7 yr with a significance $> 3\sigma$ for PKS 2155-304, presented within the previous studies (see, e.g., [153]) but concluded to be non-periodic $\gamma$-ray variability by [146]. The later analysis of the optical *R*- and LAT-band data by [150] did not yield a firm detection of the quasiperiodic variability. Later, such a variability with a period of $\sim$1.7 yr was reported by [151] with a significance of $2.5\sigma$–$5\sigma$ by different periodicity searching technique from the LAT observations of PKS 2155-304 performed during 2008–2020. However, the period of $\sim$3.4 yr was also obtained from the same data

train by means of the phase dispersion minimization (PDM, [154]) method. The authors of [152] did not find a significant periodicity after performing a trial correction.

Moreover, Refs. [155,156] reported quasi-periodicity detections for Mrk 421 and Mrk 501. In addition to the issues related to the analysis of the *Fermi*-LAT data (e.g., the energy range of 0.1–300 GeV instead of 0.3–300 GeV generally adopted for HBLs; see [157]), the reported periods (285 and 330 days, respectively) are detected below the $3\sigma$ significance and/or show some changes with time. Consequently, such detections are not robust (according to the criteria of [147]). No significant periodicity was found with a trial correction made by [152].

*3.2. Relativistic Shocks and Fermi-I Process*

Relativistic shocks are naturally expected in such supersonic outflows as the HBL jets. Shock fronts represent efficient sites for dissipating the bulk kinetic energy for accelerating leptons and hadrons up to the ultrarelativistic energies and produce a flux variability on various timescales [10,158]. A variety of shocks are expected: reconfinement shocks, stationary or moving shocks along the jet [5]. The simulations of [56] demonstrated that mildly-relativistic shocks in weakly magnetized jet flows produce relativistic particle acceleration. These events may result from the intermittent changes in the physical conditions in the innermost AGN area, which can saturate the jet with extremely energetic plasma having a significantly higher pressure than the steady-state jet flow and a forward shock front is formed [76].

Gamma-ray flares of HBLs sometimes show a long-term increase of the flux (weeks to a few months, expected by the shock propagation through the jet), superimposed by shorter-timescale variations (lasting several days to a few weeks; see, e.g., [19,42] for Mrk 421). Such rapid variations could be related to the excitation of the recollimation nozzle by the external perturbation (so-called recollimation shock caused by an external medium; [82]), or by the interaction between the moving shock front and local, jet-inherent inhomogeneities [76]. In order to explain a flaring behavior of Mrk 501, Ref. [34] considered the shock structures as follows: (1) a double shock system with forward and reverse shocks, and (2) a single shock along with a rarefaction wave. The presented model predicts correlated multi-band variability with some cross-band time lags and spectral hysteresis patterns. The average SED of Mrk 501 from the period 2009 March–August, well described by [52] within the standard one-zone SSC model, in which the bulk of the energy was generated within a single emission zone associated with the relativistic, proton-mediated shocks. The multi-zone SSC model of [57] (see Section 2.2) also incorporated a shock acceleration of electrons by a standing or propagating shock in a collimated jet. By assuming that the radio-to-X-ray flux from HBLs is synchrotron radiation of isotropically distributed electrons in the randomly oriented jet magnetic field, Ref. [159] obtained the underlying EED and adopted it to construct the SSC SED as a function of the Doppler factor, magnetic field strength and variability timescale. This method was adopted to model the VHE spectra of PKS 2155-304 and Mrk 421 during the giant outburst on 28–30 July 2006 and during the 2001 March flare, respectively. Temporal variability was assumed to result from Fermi-I mechanism, adiabatic expansion, and radiative cooling.

In HBL jets, one could expect the simultaneous existence of the perturbations having different origin and producing the erratic behavior exhibited by the observed $\gamma$-ray light curves of HBLs. Shock interaction with the turbulent jet medium (characterized by the enhanced density and magnetic field) can generate rapid fluctuations with the observed timescale $\Delta t_{obs} = (l_{inh}/V_{sh})(1+z)/\delta$, with $l_{inh}$ to be the inhomogeneity length; $V_{sh}$, the shock speed [56,76]. The observed peak flux and variability amplitude can be strongly enhanced by relativistic effects if the angle between the jet and our line-of-sight is small [76].

The collision between the fast and slower shock fronts (or high-energy plasma blobs) can trigger a system of forward and reverse shocks, which confine two subsequent emission zones and yield complex flare profiles, e.g., double-peaked flares [76,160]. According to [82],

the interaction of jet matter with an obstacle can also trigger a double-shock structure depending on the relative momentum fluxes carried by the jet and the obstacle, respectively.

The dominant particle acceleration mechanisms at mildly and non-relativistic shocks are *diffusive shock acceleration* (DSA; [161]) and *shock drift acceleration* (SDA, [162]), which are collectively referred to as *first-order Fermi acceleration* (hereinafter, Fermi-I mechanism; [56]). In DSA, the energization of charged particles is owing to the repeated shock crossings when they interact quasi-elastically with self-generated small length-scale magnetic fluctuations, which are anchored in the converging upstream and downstream plasmas and producing a magnetohydrodynamical (MHD) turbulence [56,163]. That is, energetic particles are confined in the vicinity of the shock front by their scattering on the magnetic turbulence which, in turn, is amplified by these particles; diffusively transported back and forth across the shock, each time achieving an average energy gain $\overline{\Delta E} \sim (\Delta u/c) E \propto [(r-1)/r](u_{\mathrm{sh}}/c)$ each cycle, where $\Delta u$ is the relative velocity between the shock upstream and downstream medium, $u_{\mathrm{sh}}$—the shock speed in the frame of the upstream medium, and $r$—the shock compression ratio, i.e., downstream-to-upstream fluid density ratio (of the order of 4 for strong very supersonic shocks [10]). However, the Fermi-I mechanism will not be efficient in a cold, highly magnetized relativistic plasma dominated by the Poynting flux [164,165].

Fermi acceleration can be efficient and very fast with the acceleration rates $\dot{\gamma}$ of the order of the gyrofrequency $\omega_{\mathrm{g}} = eB/mc$. This is due to the gyroresonance-dominated interactions of electrons with the MHD turbulence [10]. In the limit of Bohm diffusion (see below), the mean free paths of accelerating electrons nearly equal to their gyroradii and by achieving the highest possible energy, they can radiate synchrotron photons up to energy $\sim 150\eta(v_s/c)^2)$ MeV, with $v_s$—the shock velocity and $\eta(\leqslant 1)$—the inverse of the between the diffusion coefficient and its value in the Bohm limit [163].

For example, Ref. [166] presented an one-zone SSC model where a jet blob, containing the separate acceleration and emission zones, is moving relativistically toward us with Doppler factor $\delta$. The acceleration zone (dominated by the DSA mechanism) is represented by a slab containing shock front and is spatially separated from the emission zone. For the stationary emission, the number of electrons injected in the AZ and that of escaped into the emission zone is equal, while X-ray-to-TeV flare was modeled by time variations in the acceleration timescale, yielding more energetic electrons within shorter time intervals leading to the hardening of the $\gamma$-ray spectrum.

Ref. [167] presented a time-dependent two-zone SSC model for the MWL observations of Mrk 421 in March 2001, where the second component is (i) pre-existing and co-spatial and participates in the evolution of the active region ("background"), or (ii) spatially separated and independent, only diluting the observed variability ("foreground"). The flux variability was ascribed to the injection of relativistic electrons in the emission zone as a shock front crosses this jet area. However, a quadratic relation between the X-ray and TeV flux variabilities was not reproduced. The authors [168] also adopted a two-zone SSC model for the different MWL states of Mrk 501 during 2011, where the $\gamma$-ray emission is produced within two jet blobs containing ultrarelativistic electrons accelerated by the Fermi-I process.

According to [169], the EED established within the Fermi-I process at the relativistic shock front can be represented by a simple power law $N(\gamma) \sim (\gamma/\gamma_0)^{-s+1}$, with the EED spectral index $s = -\log p/\log \epsilon$; $p$, the probability that electron will undergoes the acceleration step $i$ (characterized by the energy gain $\epsilon$, assumed to be independent from the electron energy as $\gamma_i = \epsilon\gamma_{i-1}$). A log-parabolic EED can be established when the condition that $p$ is energy-independent is broken and the probability of the particle's further acceleration is declining as energy increases, i.e., the probability $p_{\mathrm{i}}$ of further acceleration at the step $i$ is given by $p_{\mathrm{i}} = g/\gamma_{\mathrm{i}}^q$, with $g$ and $q$ to be constants. If $q > 0$, the probability $p_{\mathrm{i}}$ decreases with energy (the so-called *energy-dependent acceleration probability* (EDAP) process).

In the case of relativistic shocks, different physical factors (the lifetime of the shock front and spatial extent) can limit the energy to be attained by charge during the Fermi-I

process. However, acceleration will eventually cease even in the absence of these factors: when the radiative energy losses (syncrotron plus SSC, inevitably associated with the acceleration) overwhelm the energy gains obtained upon the shock crossings [163]. Moreover, the microphysics of the jet turbulence represents an important factor which determines the value of the power-law photon index and the number of the energy orders passed by particles during the DSA process [28]: when electrons undergo infrequent large-angle scatterings, they produce harder power laws (than in the case of small-angle scatterings) and pass significantly more energy orders before establishing a power-law EED.

Initially, the magnetic field is thought to be random in the jet emission region, but the shock passage can compress it and produce an ordered component [170]. Generally, one assumes during the jet modeling that shocks have a direction transverse to the flow. However, the VLBI observations sometimes show features indicating that shock fronts should be oblique with respect to the jet axis and, consequently, the presence of conical shocks were suggested (e.g., [171]). On some occasions, $\gamma$-ray flares can be triggered not by propagating shocks: these events are expected also during the encounter of propagating particle density or magnetic field enhancement and stationary jet inhomogeneity (e.g., a recollimation shock; [170]).

Oblique, relativistic shocks are referred to as "superluminal", implying that they cannot be the sites of first-order Fermi acceleration [162]. As an alternative mechanism for particle acceleration is the shock-drift mechanism. In this scenario, particles can be accelerated by means of a single shock crossing when they undergo a drift parallel (or anti-parallel) to the electric field ([162] and references therein). In turn, this field can be induced when the charged particle is moving towards the shock. This mechanism is mentioned as fast Fermi process: particles are allowed to boost their energy by an order of magnitude even during a single shock encounter [5]. Note that the level of the MHD turbulence should be relatively low for a shock-drift acceleration to be the most efficient (in contrast to DSA and stochastic processes). In the case of weak turbulence, shock-drift acceleration can become dominant in oblique shocks and produce a hard-spectrum EED up to the highest energies (as obtained within the Monte Carlo simulations; see, e.g., [164]). Consequently, the Fermi-I mechanism can produce high-energy electron populations characterized by a large range of power-law indices: from very steep indices down to very hard ones ($p \simeq 1$) depending on the properties of magnetic field and turbulence, shock speed and obliquity in the case of mildly relativistic shocks [28]. Consequently, DSA and SDA complete each other in point of the acceleration capability; the first mechanism is dominating in the case of strong turbulence near the shock front, while the SDA is more efficient when the magnetic field is substantially more laminar on larger spatial scales [28]. This model was adopted by [56] to reproduce the broadband SED and $\gamma$-ray light curves presented in [52].

In the case of ultrarelativistic shocks (with Lorentz factor $\Gamma_{sh} \gg 1$), only the particles with $\gamma \gg \Gamma_{sh}$ can manage to cross the shock front from downstream to upstream. In a magnetized medium, such crossing is possible only for parallel or quasi-parallel shocks, in the case of small angles between the magnetic field and the flow direction [28,158]. Consequently, the Fermi-I processes is expected to be much less efficient (especially, in the presence of perpendicular and quasi-perpendicular shocks), and the EED spectral indices are significantly softer than in the case of mildly-relativistic shocks, with a universal value $\sigma \approx 2.2$–2.3 [172].

Ref. [76] modeled a rapid MWL variability by assuming that the $\gamma$-ray emission is produced via the SSC mechanism and accounting for (i) the energy stratification established by particle acceleration at shock fronts; (ii) electron cooling by synchrotron emission; and (iii) the effects of light-travel delays for the synchrotron emission providing the seed photons for the IC up-scattering. An MWL flare was produced by the collision between the relativistic shock and jet inhomogeneity, triggering a forward-reverse shock structure. These simulations indicated that relative delays between the $\gamma$-ray and synchrotron flares are determined by the energy stratification and geometry of the emitting regions confined between the forward and reverse shocks and yielding both negative and positive time

delays depending on the spectral band. Moreover, the light-travel effects related to the seed photons for the EIC upscatter may lead to the delay of the $\gamma$-ray variability with respect to those observed at synchrotron frequencies when the jet axis is (nearly) aligned with our line-of-sight.

The 2–8 keV polarimetric observations of Mrk 421 and Mrk 501 with IXPE showed that the X-ray polarization degree was more than a factor of two–three higher than the optical one. These results were explained by the shock presence in the emission zone: higher-energy, hard X-ray emitting particles should populate the magnetically more-ordered region closer to the shock front, and then diffuse away to the area with less-ordered magnetic field, producing optical emission with lower polarization [70,71].

In order to explain the very hard VHE spectra of UHBLs, Ref. [82] revisited the one-zone model by assuming that electrons are co-accelerated with protons by relativistic internal or recollimation shocks in the case of the physical situations as follows: (1) low jet magnetisation and (2) electrons could be preheated in the shock transition layer, yielding large minimum Lorentz factors when involved in the Fermi-I process. While acceleration by a single shock was sufficient for the hardest UHBL SED, re-acceleration on a second shock was considered. The $\gamma$-ray emission from the accelerated proton population (with the same number density as the electrons) did not make a significant contribution.

*3.3. Jets-in-Jet Model and Relativistic Magnetic Reconnection*

In the framework of a jets-in-jet model (e.g., [133,137]), the TeV-band emission is generated in the small-size emitting regions which move relativistically with respect to the main jet. In turn, the latter also is relativistic characterized by the overall bulk Lorentz factor $\Gamma_b$. It was concluded that such a geometry is capable for producing a high-amplitude variability on timescales which are significantly shorter than the light-crossing time of the central SMBH. Namely, the emission from a tiny source zone to be beamed through a narrow cone and the observed TeV-band flux is amplified without needing to impose any extreme requirement on the emitting zone.

Moreover, detections of extremely fast flaring TeV emission in Mrk 501 and PKS 2155-304 (see above) impose limits on the spatial scales of the high-energy emission region, which are much shorter than the light-crossing time of the central SMBH (amounting to hours for the blazars with the SMBH masses $\sim 10^9 M_\odot$) and, consequently, it is reasonable to suggest a compact jet emission region to be the source of such TeV flares. On the other hand, the escape of TeV photons from such very compact emission zone implies that the latter should move with a bulk Lorentz factor $\Gamma_{em} \gtrsim 50$ (in order to avoid the annihilation within the soft radiation fields; [133]).

Magnetic reconnection is considered as very efficient and rapid mechanism for using the jet magnetic field energy for accelerating electrons to the energies required, e.g., to upscatter synchrotron photons to gamma-rays. Magnetic field lines in the HBL jets may undergo breaking and reconnection. Consequently, a significant portion of the magnetic energy can be converted into the kinetic energy of jet plasma and accelerate particles. In the particular medium, the time evolution of the magnetic field is given by $\delta \mathbf{B}/\delta t = \nabla \times (u \times \mathbf{B}) - \nabla \times \eta(\nabla \times \mathbf{B})$, where $\eta$ is a magnetic diffusivity of the medium. In astrophysical plasmas with high magnetic conductivity, the first term on the equations right hand side is generally dominant and, consequently, magnetic field is frozen-in and no reconnection can occur. However, this term may become negligible in some area of the jet plasma, e.g., around the stagnation points, over some lines or surfaces characterized by $u \simeq 0$ [5]. Thereby, strong currents or current sheets with non-zero electric field are induced, leading to the plasma heating and particle acceleration. The relativistic (MHD) and three-dimensional particle-in-cell (PIC) simulations demonstrated that a blazar jet can become unstable to the kink-mode instabilities [173], producing a filamentary current density pattern which is inclined to the magnetic reconnection. Namely, such currents may trigger growing kink instabilities and turbulences which can then lead to the development of an anomalous resistivity. The latter can strongly amplify the magnetic field's dissipation [174].

While the relativistic shocks convert a fraction of the jet kinetic energy of the jet, magnetic reconnection is a highly efficient mechanism for extracting a magnetic field energy and using it for the particle acceleration to ultrarelativistic energies. Namely, this process can rapidly convert a sizeable magnetic energy into the particle kinetic energy via the rearrangement of the field lines [175]. The simulations showed an spontaneous appearance of plasmoids (or magnetic flux tubes) in the sufficiently long and thin current layers, owing to the tearing instability [176]. These plasmoids enhance the overall reconnection rate by trapping the energised particles and evacuating them along with the reconnected magnetic field from the so-called magnetic X-points. They can represent the compact blobs adopted in the different emission models [176].

The reconnection-based mini-jet model was proposed to explain the extremely fast variability shown by the TeV-detected blazars [137]. The model incorporates two wedge-shape regions with relativistically flowing plasma ("mini-jets") and separated by a stationary shock. In such a geometry, mini-jets are perpendicular to the relativistic axis. They are formed in the process of magnetic reconnection and leave the reconnection site in the form of blobs which are moving with relativistic speeds. They produce a fastly variably TeV-band emission within a narrow beam by means of the SSC mechanism (Figure 3). The sequence of the fast TeV-band flares shown by PKS 2155−304 in 2006 July was explained by the existence of the multiple, reconnection-born mini-jets [137].

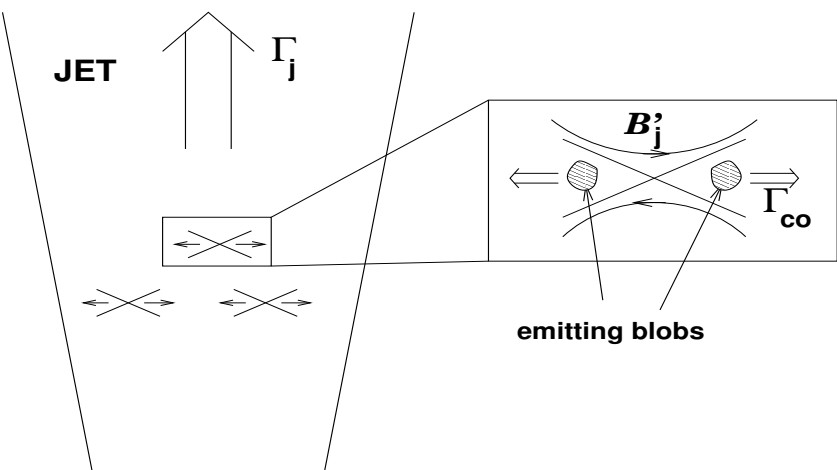

**Figure 3.** A schematic representation of the jets-in-jet geometry. Reproduced according to the MNRAS guidelines from [137].

Ref. [175] showed that the magnetic reconnection process is often relativistic in the high-energy universe: the energy density of the reconnecting magnetic field $B_0$ is higher than that of the ambient medium: $\sigma_0 = B_0^2/(4\pi w_0) \gg 1$, with $w_0$, the relativistic enthalpy including the rest-mass energy. In such a situation, a relativistic magnetic reconnection is the most efficient mechanism to dissipate the magnetic field energy and accelerate particles. In that case, a hard power-law particle energy distribution $N(\gamma \propto \gamma^{-p})$ can be established with $p \rightarrow 1$ [177].

Relativistic magnetic reconnection has been studied for blazar jets having an electron–ion or mixed composition. For example, Ref. [138] performed the 2-D PIC simulations of this process for the electron–positron and electron–proton compositions. It was concluded that the reconnection mechanism yields (i) efficient conversion of the magnetic energy into that of accelerated particles; (ii) an extended, non-thermal relativistic energy distribution of particles, and (iii) plasmoids characterized by a rough equipartition between the energies of magnetic fields and that of the relativistic particles. Ref. [178] demonstrated that fast magnetic reconnection can form a self-similar chain of plasmoids which grow in time, while their interiors undergo an compression and amplification of the internal magnetic fields. Consequently, particle energization follows from the conservation of magnetic

moment. If particles are injected into plasmoids with a power-law energy distribution, the aforementioned process conserves the original functional shape but adds a nonthermal tail described by $f(E) \propto E^{-3}$ at higher energies, followed by an exponential cutoff with the maximum energy increasing with time as $E_{\mathrm{cut}} \propto \sqrt{t}$.

Ref. [179] presented a model for flares produced by individual reconnection plasmoids. In this model, the peak luminosity and flux doubling time-scale were represented as the functions of the plasmoid size and momentum. Ref. [180] interpreted the exceptional X-ray outburst in 2013 April 11–19 and simultaneous MWL behavior in the framework of magnetic reconnection scenario. Here, the multi-hour flux variability is modeled as a combination of the emission from the plasmoids of different size and velocity. As for the sub-hour variability, one a adopted a scenario incorporating a dominant emission from a single small plasmoid which is moving across the magnetic reconnection layer.

An uncorrelated VHE flare of Mrk 421 observed on 4 February 2017, and explained within two-zone SSC scenario by the existence of a compact second blob containing highly energetic electrons with a narrow range of Lorentz factors (see Section 2.2) was suggested to result also from a magnetic reconnection [74]: the blobs containing ultrarelativistic particles could be formed at the jet's reconnection sites and produce a high-energy emission. During the reconnection process, the dissipated magnetic energy is converted into the kinetic energy of nonthermal particles, leading to a decrease in the magnetic field strength with increasing gamma-ray activity. In turn, the ratio $U_B/U_e \sim 10^{-3}$ can be obtained, which is needed to reproduce the observed broadband SED.

One of the explanations of the resent X-ray and optical polarimetric results obtained for Mrk 421 and Mrk 501 was related to the turbulence-induced reconnection in the jet characterized by transverse velocity gradients and, therefore, yielding higher-ordered fields in the jet's transverse direction [70,71].

*3.4. Jet Turbulence and Fermi-II Process*

Magnetized turbulence is very important for blazar jets in different aspects (see [181] for a review): (a) at the least, it provides scattering agents for DSA; (2) turbulence generates magnetic reconnection, or the converse; (3) it represents an efficient mechanism of particle acceleration, by means of the stochastic or second-order Fermi (Fermi-II) acceleration in a shock downstream region: a particle interacting with randomly moving magnetic inhomogeneities with a typical velocity dispersion $\beta_{\mathrm{m}} c$ can gain a large energy stochastically with a rate $\propto (\beta_{\mathrm{m}} c)^2$.

As the blazar jets propagate, its interactions with the ambient medium can lead to the different instabilities and mass loading. Consequently, the turbulence responsible for the Fermi-II process can be triggered by (i) a Kelvin–Helmholtz instability (see, e.g., [139]); (ii) a current-driven instability [182]; (iii) a recollimation shock [83].

Refs. [70,72] explained a large change in the polarization degree from X-ray to optical frequencies as follows: in the jet plasma crossing a shock front and having a turbulent magnetic field, particle acceleration is expected to be the most efficient in those cells where the magnetic field is nearly parallel to the shock normal. Consequently, a higher polarization degree and stronger variability should be observed at higher frequencies.

Ref. [183] presented a relativistic turbulence model for the very fast TeV variability of PKS 2155-304, in which a MHD turbulence in the blazar jet generates compact plasma blobs on the spatial scales smaller than the event horizon radius of the central SMBH (similar to the jets-in-jet scenario). These sub-regions move relativistically in random directions and the variability time-scale is determined by the size of each region in their own comoving frames. In the case of the variable orientation during the turbulent blob movement, the observer may receive its radiation only during the short time interval when the beam is pointed to the Earth.

In order to achieve a satisfactory representation of the very hard VHE spectrum of the UHBL source 1ES 1101$-$232, Ref. [184] used a time-dependent SSC model where extremely hard electron distribution is achieved within the stochastic acceleration yielding a steady-

state, relativistic, Maxwellian-type particle distribution peaking at high electron Lorentz factors $\sim 10^5$. This distribution represents a time-dependent solution of the Fokker-Plank equation that incorporates the radiative energy losses of accelerating particles and is capable to reproduce the observed hard TeV-band spectra. Depending on the physical conditions in the jet emission zone, (e.g., if particles undergo cooling beyond the acceleration zone, or the jet medium is clumpy), the combination of different pile-up distributions is capable of interpreting the observed features.

The uncorrelated VHE flare of Mrk 421 observed on 4 February 2017 was explained within the two-zone SSC scenario, incorporating the presence of a compact second blob of highly-energetic electrons with a narrow range of Lorentz factors (see Section 2.2). This event was also explained in the framework of the Fermi-II acceleration: quasi-Maxwellian EEDs could be established in the process of energy exchanges with resonant Alfven waves in a highly turbulent medium [74].

The broadband SED of Mrk 501, constructed by using the MWL data collected on 9 June 2012, showed a transient UHBL nature with the higher-energy peak at $\sim 2$ TeV [185]. A two-zone model was adopted for this case: the first, larger zone dominating in the optical and MeV energy ranges; to be steady or slowly variable. The second, smaller zone, spatially separated from the first one and characterized by a very narrow EED (owing to the stochastic acceleration), was the dominant source of the variable X-rays and VHE emissions, producing also the aforementioned TeV-band high-energy.

Moreover, Ref. [186] reported hard high-energy spectra characterized by the photon index $\Gamma < 1.5$ (down to $0.89 \pm 0.29$) above 10 GeV on 17 occasions from the *Fermi*-LAT 7-yr data of Mrk 501, each with 30-d integration time. The corresponding SEDs (whenever the VHE spectral points were available) were modeled by using a two-zone SSC scenario: two co-moving blobs (with $\delta \sim 10$) characterized by the narrow power-law and relativistic Maxwellian EEDs, established by means of the first and second-order Fermi mechanisms, respectively. We also found a number of the instances of very hard LAT-band photon index for Mrk 501, as well as for another HBL source 1ES 0033+595 ([69,187]; see also Figure 4). Note that these objects showed the features of the efficient stochastic acceleration, as well as very fast X-ray variability explained by presence of small scale jet inhomogeneities with strong turbulent magnetic fields (see [68,187]).

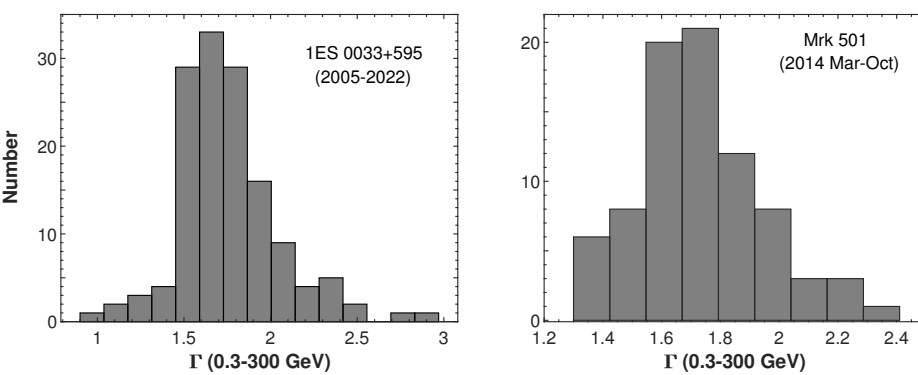

**Figure 4.** Distribution of the LAT-band photon indices in the HBL sources 1ES 0033+595 (**left**) and Mrk 501 (**right**). The histograms are constructed using the values derived by us from the LAT data analysis according to the recipe provided in [187].

The 2-D MHD modellings of a mildly relativistic shock propagating through a jet inhomogeneous medium showed that the post-shock jet regions may become highly turbulent if there are pre-shock density inhomogeneities. Moreover, magnetic fields can be strongly amplified in these regions due to the stretching and folding of field lines existing in the turbulent velocity field [140,188]: if the initial magnetic field is perpendicular to the shock normal, it will be compressed by the shock front and then undergo an additional

amplification by turbulent motions. The amplified magnetic field evolves into a filamentary structure and the turbulence spectrum is flatter than the Kolmogorov function (see below).

Charged particles in turbulent jet plasma are expected to be accelerated via interactions mainly with Alfven waves propagating in the magnetized medium [189]. The astrophysical collisionless turbulence is generally represented as an energy cascade which is spanned over the different orders of spatial scales (from large down to small wavelengths). Generally, most of the fluctuation power (in velocity and electromagnetic fields) are carried by the larger-scales turbulence[181]. The wave energy distribution and intensity are given by $W(k) = (\delta B k_0^2/8\pi)(k/k_0)^{-q}$ and $I_k = I_0(k_0/k)^q$, respectively, with $k = 2\pi/\lambda$ to be the wave number; $\delta B$, the turbulent component of the jet magnetic field; $q$, the turbulent field spectral slope: $q = 3/2$, $q = 5/3$ and $q = 2$ for the Kraichnan, Kolmogorov and 'hard-sphere' turbulences, respectively. For the wavenumbers below an inverse correlation length $k_0$, the wave intensity per logarithmic bandwidth is assumed to be equal to the background field intensity; i.e., $I(k) = B_0^2 k^{-1}$ when $k < k_0$ [190].

The Fermi-II process is based on (quasi)elastic reflections (scatterings) of the charged particles by the magnetic inhomogeneities or plasma waves. Consequently, if there are waves propagating towards both directions at a given position, a stochastic acceleration of charged particles can be developed [190,191]: particle gain or lose energy when the "mirror" is approaching or receding, respectively. However, the simulations showed a higher probability of the head-on collisions compared to the rear-on reflections and, on average, it can gain energy. Note that the energy gain per bounce is proportional to the square of the mirror velocity (hence the name of the mechanism: second-order Fermi acceleration). However, the net energy gain depends also on the scattering rate. The Fermi-II mechanism is widely accepted to be a stochastic process and is known also as stochastic acceleration. This process becomes less efficient at the energies $\gamma > \gamma_0 = \Omega_{e,0}/k_0 c \gg 1$, with $\Omega_{e,0} = eB/m_e c$; $k_0$, the inverse correlation length.

Two different types of the stochastic particle transport are considered [192]: (1) in the case of the small spatial scales, charged particles interact with the MHD waves moving in the local magnetic field and undergo a stochastic acceleration. The mean-free path of particles on such scales is equal to the coherence length for the MHD turbulence, $l_{MHD} = c\sigma_{mag}/(3D)$, where $\sigma_{mag} = (V_A/c)^2$; (ii) particle transport is implemented via diffusion in the case of large spatial scales. In such a situation, a mean free path is determined by the relativistic electron's Larmor radius, $r_L = E/qB$ with $E$ and $q$, electron's energy and charge, respectively, and $B$, the blob's magnetic field. This regime is referred as a Bohm diffusion. When the electron energy becomes sufficiently high, its Larmor radius becomes comparable to the blob radius and this high-energy electron can escape from the system. Therefore, no further acceleration is possible in the case of $r > r_L$ (so-called Hillas condition [193]).

Therefore, a stochastic acceleration can be characterized by a diffusion coefficient in the momentum space. The magnitude and scaling of this coefficient affect the evolution of the EED in the Fokker-Planck equation [194]. In most astrophysical situations, one considers a large-amplitude turbulence, since the stochastic acceleration is found to be very slow within in the case of small-amplitude turbulences: the acceleration timescale (generally referred as the time required on average to double a particle energy), is proportional to $(\delta B/B)^{-2}$. In the case of strong turbulence, particularly important is the relativistic limit where the Alfven speed $v_A \sim c$, where $v_A = B_0 c/\sqrt{4\pi h n + B_0^2}$; $h = (\rho + P)/n$, $n$, $\rho$, and $P$ are the specific enthalpy, number density, total energy density, and gas pressure (measured in the local plasma frame; [195]). Therefore, we are mainly interested in those particles interacting with large-scale (the scattering timescale is a growing function of energy) modes of a fast-moving turbulence spectrum [181].

The simulations of [190] showed that the efficiency of the Fermi-II process can be comparable to that of the shock acceleration. Moreover, it operates over much longer timescales than the Fermi-I mechanism. Stochastic acceleration, on the other hand, may be present on some level in the turbulent downstream of shocks and deliver pre-accelerated particles

to the shock front (see, e.g., [26]). Moreover, there can be a combined acceleration process: firstly, particles are efficiently accelerated at the shock front via the Fermi-I mechanism and after the escape into the shock downstream region, they will be involved in the Fermi-II process. Consequently, their energy can be boosted sufficiently to allow particles to return in the shock acceleration zone and repeat the previous acceleration cycle [191].

In the local jet frame, the mean free path represents the spatial scale at which the particle's momentum vector is deflected by $\pi/2$ on average [10]: the wave-particle interaction can be presented by the scattering relation $\lambda_\parallel = \lambda_1(\rho_1/\rho)(r_g/r_{g1})^\alpha \equiv \eta_1 r_{g1}(p/p_1)^\alpha$, $\kappa_\parallel = \lambda_\parallel v/3$, with $\lambda_\parallel$ ($\kappa_\parallel$) is the mean free path in the local frame and parallel to the field $\mathbf{B}$, $v = p/m$—particle's velocity in the local frame, $r_g = pc/(QeB)$—gyroradius of a particle carrying a charge $Qe$, $\rho$—the plasma density with a far shock upstream value of $\rho_1$. Note that the condition $\lambda_\parallel \gtrsim r_g$ is the so-called Bohm limit, representing a fundamental bound for the physically meaningful diffusion. The parameter $\eta \equiv \lambda_\parallel/r_g \propto p^{\alpha-1}$ characterizes the scattering strength and, hence, the importance of particle's cross-field diffusion: when $\eta \sim 1$ (i.e., $\lambda \sim r_g$) at the Bohm diffusion limit, $\kappa_\perp \sim \kappa_\parallel$ and particles diffuse across magnetic field lines quickly. Note that the Bohm diffusion corresponds to extremely turbulent magnetic fields with fluctuations satisfying $\delta B/B \sim 1$ and $\alpha = 1$. The condition $\lambda \geqslant r_g$ is required for the physically meaningful diffusion resulting from gyroresonant wave–particle interactions. Therefore, the case $\alpha = 1$ is highly important for the different astrophysical situations [10].

The $\gamma$-ray SED of Mrk 501 corresponding to the 1–5 May 2009 window was modeled within the one-zone SSC scenario yielding $\alpha = 1.5$, implying the interactions with weaker turbulence for more energetic particles which undergo a diffusion on larger spatial scales [10]. The modelling indicated that the turbulence strength declines with distance from the shock front in the relativistic jet medium. Consequently, the particle diffusion becomes significantly different from the Bohm limit at all energies, and the diffusion scale $\lambda$ will increase with the particle momentum. This is required for leptonic models in order to explain very hard HE and VHE spectra, which yielded $\lambda_\parallel \propto p^\alpha$ with $\alpha \gtrsim 1.5$, associated with a weaker turbulence. It was concluded the the electron mean free paths should be orders of one magnitude larger than their gyroradii at the Lorentz factors derived from the SED simulations [10].

The stochastic acceleration rate depends on the wave spectrum [190] as follows: for $q = 2$ (hard-sphere turbulence), charged particles have the same, rigidity- and energy-independent mean free paths and, therefore, the Fermi-II mechanism accelerates them at a constant rate. The situation is different within the Kolmogorov turbulence ($q = 5/3$): the mean free paths and acceleration rate of particles decline while the energy increases, and the Fermi-II process becomes gradually inefficient with higher-energy particle distributions. As the parameter $q$ increases and the turbulence spectrum becomes steeper, a larger portion of the turbulence energy is contained in longer waves which, in turn, can interact resonantly with higher-energy particles. Consequently, steeper-spectrum turbulences are capable for producing harder particle distributions that the Kolmogorov turbulence. Consequently, hard-sphere scattering centers are more efficient to accelerating charged particles compared with a Kolmogorov-type wave ensemble. On the other hand, the latter is more efficient compared to the Kraichnan spectrum ($q = 3/2$). The spatial scales for the Kolmogorov turbulence are up to an order of magnitude shorter than in the case of the hard-sphere spectrum [10]. It was concluded that a hardening in the turbulence spectrum shifts the EED cutoffs to higher energies [190].

The EED behavior during the stochastic acceleration depends on the strength of the background magnetic field: the intensity ratio of the Alfven waves in the shock downstream region to those in the upward region is a function of the quasi-Newtonian Alfvenic Mach number $M = u_1/u_{A,1}$, with the shock proper speed $u_1 = c\sqrt{\Gamma_1^2 - 1}$; $\Gamma_1^2$, the Lorentz factor of the upstream bulk flow; $u_{A,1}$, Alfven speed in the shock upstream region. In the case of relativistic shocks, the Alfven waves are seen to propagate predominantly backward for relatively low Mach number shocks [196]. Consequently, the compression ratio $r_k$ of the scattering center becomes larger than the gas compression ratio $r$ and, eventually, a

significantly harder EED is established (compared to the frozen-in case; [190]). In the cases of weak magnetic fields and a quasi-Newtonian Alfvenic Mach numbers much exceeding the critical Mach number ($M \gg M_c = \sqrt{r}$; with $r$, the shock compression ratio), the effects of stochastic acceleration are overwhelmed by the much stronger Fermi-I acceleration [190].

Ref. [190] found that the contribution of the Fermi-II mechanism to the particle energy distribution is insignificant compared to that of the Fermi-I acceleration at the shock for high Alfvenic Mach numbers ($M = 1000$, corresponding to $B_0 \approx 1.4$ G in a Hydrogen plasma); the distribution sustains its shape and energy range unchanged at least for tens of thousands of the electron's mean free paths, regardless of the applied turbulence spectrum. For the stronger magnetic fields ($M = 10$ and $M = 3$, corresponding to $0.14$ G and $0.46$ G, in a Hydrogen plasma and to $4.6$ and $15$ mG in a electron-positron plasma, respectively), the stochastic acceleration effects are much more significant: the Fermi-II mechanism will further energize particles just after the shock front, and the entire energy spectrum gradually shifts to the higher energies [190].

When the particle energy becomes higher than the turnover energy $\gamma_0$, the rate of energization is expected to go down: after this threshold, the particle's mean free path will increase much faster, leading to the decrease in the stochastic acceleration efficiency. Moreover, particles will be able to escape into the shock downstream region and even manage to return back to the shock (at least for $M = 3$). After the turnover energy, particles undergo a pile-up and, consequently, one expects appearance of the narrow distribution bump immediately beyond the $\gamma_0$ [190]. Note that one of the alternative scenarios to explain the narrow spectral feature at VHE is pileup in the high-energy range of the relativistic EED due to stochastic acceleration [54].

Mrk 501 showed peculiar high-energy characteristics of during the LAT observations in 2009 May, exhibiting a flaring activity and spectral hardening above 10 GeV while a weak activity was detected at the lower energies. In order to explain this behavior, Ref. [197] adopted a "leading blob" model: the observed radiation was produced within several emitting blobs, where electrons were accelerated to relativistic energies by the Fermi-II mechanism and produced a narrow (piled-up) distribution. All blobs were assumed to have similar physical parameters, with exclusion of the characteristic energies of their EEDs. A TeV-band flare and hard spectral feature was reproduced by dominance of one (or a few) of the radiating components. This so-called leading blob could boost its apparent luminosity by changing its Doppler factor or the injected energy.

Although the spatial scales of stochastic acceleration are enormous compared to the Fermi-I process, they are still orders of magnitude smaller than the spatial scales resolvable by the current VLBI observations [190]. The acceleration timescales can be also very short: the time required to shift the entire EED from the initial energy range to the turnover energy takes from 10 to 50 min in the $M = 10$ case, and for $M = 3$ the acceleration times are $\lesssim 1$ min in the shock frame.

In the process of the interaction between the HBL jets and the ambient medium, a sharp boundary layer may be formed in the case of the large velocity difference between them. Moreover, a jet may have unequal bulk Lorentz factors at different distances from the axis [5]. Rayleigh–Taylor-type instabilities can be triggered at such spine-sheath interfaces in the shearing layer of the thickness $\sim 0.1 R_{\text{jet}}$, where the resulting turbulence can accelerate particles via the Fermi-II mechanism (see, e.g., [198]). Namely, different simulations showed that this process can accelerate electrons up to PeV energies and protons up to EeV energies [5]. The average energy gain per interaction $\overline{\Delta E} \propto (u/c)^2 E \propto (l/c)^2 (\delta V_z/\delta x)^2 E)$, with $V_z$ to be a flow speed along the z-axis (transverse to the x-axis directed along the spine-sheath interface); $u = l(\delta V_z/\delta x)$, the speed change of the scattering centers in the particle's frame, crossing the shearing flow along the x-axis and having the mean free path $l$. The acceleration time $\tau_{\text{acc}} \propto (\delta V_z/\delta x)^2$, i.e. a shear acceleration is faster for those particles which already have high energies. Moreover, this acceleration is more efficient for protons with large $l$ than for electrons [135].

### 3.5. Jet–Star Interactions

Ref. [199] proposed a possibility of fast TeV-flares by compact magnetized blobs, produced when red giant (RG) stars cross the blazar jet close to the central SMBH. In the framework of jet–star interaction, the RG stars should cross the jet at different distances from its base. In this process, one expects a shocks trigger which will transfer the bulk kinetic energy to charged particles [200]. Stars are expected to traverse faster the innermost jet parts which are thought to be narrower than those with large separations from the central engine and, therefore, are the most plausible sites of the aforementioned fast flares. In the case that the RG is even slightly tidally disrupted by the SMBH, there can be a large amount of stellar material to be blown by the jet. This material ("bubble") will expand quickly, until being shocked by the jet ram pressure [201]. A shock will propagate through the bubble, heats up its material and accelerate particles to relativistic speeds [202]. The shock will propagate until reaching the stagnation radius, where the bubble and jet pressures are equal. At this position, one expects formation of a double bow shock structure, which energizes via the Fermi-I mechanism up to ultrarelativistic energies. The accelerated particles (primarily, leptons) can contribute to the jet's emission [202]. Namely, these particles are expected to radiate in $\gamma$-rays predominantly through the proton-synchrotron mechanism or EIC-upscattering by electrons (assuming the jet synchrotron photons as external). Within this scenario, the variable $\gamma$-ray emission might be produced during relatively short time interval [203]. Ref. [199] adopted this model to the minute-scale TeV flares superimposed on the longer (daily timescales) $\gamma$-ray variability of PKS 2155$-$304 in 2006 July.

According to [203], advection escape of charged particles dominates their radiation cooling during the star interaction with a moderately powerful jet (as accepted to be the case for HBLs). The produced radiation was found to peak from X-rays to MeV energies in the synchrotron emission (depending on the fraction of energy in magnetic field). Another peak can be situated in the 100–1000 GeV range for the IIC-upscatter, depending on the stellar type: the cooler (either older or less massive) stars are expected to yield the higher SED peak energy (up to ∼1 TeV). The radiation spectrum is related to the efficient advection of low-energy electrons even in the case relatively high magnetic fields); Interactions of jets with cold stars may yield even harder IC spectrum owing to the Klein–Nishina effects [203].

The emission generated during the jet–star interaction events can be relatively persistent at high energies, through either IC or synchrotron mechanisms within low magnetic fields (generally expected in HBLs). However, the steady state emission of the whole population seems to be undetectable [202]. Within strong magnetic fields (corresponding the equipartition value; viz., in the small-scale jet region with relativistic magnetic reconnection in HBLs), emission from the jet-star interaction can be seen at the energies ∼ 100 MeV as a bright, fast flaring instance superimposed on the persistent, lower-level IC radiation [202].

Note that [44] reported the MeV-excess SEDs from the period 2016 April–August and this result was explained by a possible jet interaction with a wind-blown bubble from a nearby red giant star.

### 3.6. Impact of Disc Instabilities on the Observed $\gamma$-ray Variability

Similar to other spectral ranges, the $\gamma$-ray variability of HBL frequently carries out a lognormal character, i.e., the $\gamma$-ray fluxes are preferentially log-normally distributed (when the observations are not limited by poor statistics; see [135] for a review). Several scenarios about the lognormality origin are proposed:

- First of all, a lognormality hints at the impact of the accretion disk instabilities on the jet [135,204]: there should be independent density fluctuations in the disk on the local viscous timescale, characterized by negligible damping. They can propagate toward the innermost disc area and couple there producing a multiplicative behavior. If the latter is transferred to the jet flow (e.g., via the jet collimation rate), the $\gamma$-ray emission can be modulated accordingly. However, the timescale for particle acceleration and radiative losses within the jet should be correspondingly small for this purpose.

Lognormal variability in the different energy range and over various timescales is then anticipated.

- Cascade-related emission processes (see Section 2.3) are also thought to lead to log-normal flux distributions [129]. However, the latter are expected only in the optical-to-$\gamma$-ray ranges. Moreover, there are limited timescales over which log-normality can be detected (i.e., from sub-hour to yearly timescales in the TeV band; [135]). Moreover, there can be some limitations by the gap travel time for the magnetospheric processes and from the dynamical or escape properties of the hadronic cascades, e.g., [135].

- Alternatively, the lognormal variability can be produced in the case of random fluctuations in the particle acceleration rate [205]. However, one should observe an energy-dependent lognormality in this case, to be progressively weakly expressed towards lower energies and disappearing beyond some threshold energy. Moreover, fluctuations in the acceleration rate can be also characterized by the Gaussian distribution of the photon indices along with the lognormal flux distribution [205].

It is possible that the lognormality in different states (long level and short term flares, low and high levels) can be dominated by one out of the aforementioned processes [135], or the observed lognormality stems from their their combination.

The lognormal $\gamma$-ray variability has been reported for a number of HBLs. Namely, this was the case for the observations performed with LAT (Mrk 421 [44,206,207]), Mrk 501 ([208]), 1ES 0033+595 ([187]), PKS 2005−489 ([209], RGB 0136+391 ([210]), H1722+119 ([210]) etc.) and Cherenkov-type telescopes (Mrk 421 [44,207], Mrk 501 [134,208] and PKS 2155−304 ([134]).

## 4. Future Prospects

The next generation of the ground-based and space telescopes are crucial for simultaneously accessing the entire $\gamma$-ray domain, carrying out the polarization measurements and studying the flux variability down to the shortest timescales. In particular,

- The Cherenkov Telescope Array Observatory (CTAO) is a next-generation IACT array, using telescopes of multiple sizes to achieve a high sensitivity in the 20 GeV–300 TeV energy range [211,212]. The observatory installations in the Southern and Northern Hemispheres will provide visibility of the entire sky and a sensitivity at least an order of magnitude higher than those of the current major Cherenkov telescopes (H.E.S.S., MAGIC, and VERITAS).

- The ASTRI ("Astrofisica con Specchi a Tecnologia Replicante Italiana") mini-array incorporates a technologically innovative solution for small size (about 4 meters diameter) and large field-of-view (more than 10 degrees) IACTs. It is sensitive in the range 1–200 TeV, achieves an angular resolution of a few arcmin and is devoted to study various types relatively bright VHE sources (a few $\times 10^{-12}$ erg cm$^{-2}$s$^{-1}$ at 10 TeV; including HBLs) at the energies beyond 10 TeV [213,214]. A prototype telescope, deployed on Mt. Etna (Italy), started its scientific operations in 2018.

- The space missions AMEGO (All-sky Medium Energy Gamma-ray Observatory) and AMEGO-X (the funded projects), will detect $\gamma$ rays through both Compton scattering and pair production, filling a "MeV gap" in sensitivity [215] . They are optimized for continuum sensitivity in the MeV range in different ways. AMEGO-X uses monolithic silicon pixel detectors for a lower energy threshold and higher low-energy effective area than AMEGO. AMEGO also has the Low-Energy Calorimeter that enhances the polarization and narrow-line sensitivity. For blazars, IC scattering is typically unpolarized or has a very low polarization degree (a few to ten percent) in a partially ordered magnetic field, while the hadronic models usually predict at least 20% polarization degree in the MeV band [216]. Consequently, one will be able to discern the underlying emission mechanism directly from observations.

- Southern Wide-field Gamma-ray Observatory (SWGO) as a next-generation Water Cherenkov Detector (WCD) instrument that will provide the observational coverage of the southern sky with nearly continuous up-time and an instantaneous field of view

(FOV) of $\sim$2 sr at energies from 100 GeV to above hundreds of TeV from a site in the Andes mountains. Simultaneous operations with CTA are planned [217].

- e-ASTROGAM is a proposed space mission for measuring $\gamma$-ray emission in the range from 300 keV to a few GeV. The e-ASTROGAM is expected to reach a sensitivity by one-two orders of magnitude higher than its predecessors, and offers enhanced capabilities to detect fast transient events in soft $\gamma$-rays [218].

These instruments are anticipated to detect much larger samples of HBLs, achieve much higher temporal/spectral coverage and angular resolution. They will be useful to search for the specific spectral features (predicted in the framework of different theories and simulations) and, especially, for the $\gamma$-ray SED components anticipated within some hadronic cascades. These instruments will also collect a large number of high-resolution datasets from the $\gamma$-ray flares of HBLs; search for (quasi)periodicities, time delays and spectral features; constrain the location, geometry and kinematics of the $\gamma$-ray emission zones; allow to select the valid emission scenarios. Thanks to the planned jump in sensitivity, a detailed exploration of very fast sub-minute variability (particularly, in the TeV-band) will be accessible that is crucially important to study the properties of the emission zones and their locations, leading to great progress in our understanding of the HBL jet physics. High-level VHE spectral analysis and long-term monitoring of the HBLs and UHBL sources will allow us to deal with the challenges associated with the current emission scenarios.

Note that the detection of UHE photons from HBLs still has not been reported by means of the current instruments; perhaps they are considered as one candidate class to produce such events [92]. For examples, Ref. [219] analyzed the data obtained for Mrk 421 and Mrk 501 with High Altitude Water Cherenkov (HAWC) Gamma-Ray Observatory during 2015 June and 2018 July. Although this array is sensitive in the 300 GeV to >100 TeV energy range (see [220]), the maximum energies at which the Mrk 421 and Mrk 501 were detected are 9 and 12 TeV, respectively [219]. Note that the detection of 12 sources of the UHE photons up to energies 1.4 PeV with the Large High Altitude Air Shower Observatory (LHAASO; [221]) was reported by [222]; none of these events are expected to have an extragalactic origin: these photons strongly interact with the EBL and their detection poses a challenge to even the next generation of the Cherenkov-type telescopes. Namely, the attenuation length to about 30 kpc around 1 PeV, which increases to the order of 10 Mpc around $10^{19}$ eV [92]. On the other hand, the expected modification of the reaction $\gamma\gamma \to e^{\pm}$ by Lorentz invariance violation (LIV, if it exists) at energies $E \gtrsim 10$ TeV can lead to the reduction of cosmic opacity. Consequently, this effect should allow photons of this extreme energy range to evade absorption and reach the Earth (see [223]).

The detection of such absorption anomalies is still very problematic: the performance of current TeV-instruments is not capable of obtaining high-quality spectra at energies higher than 10 TeV. Note that [224] did not find any signature of the LIV existence in the energy spectra of two PeV sources, and the lower limits on the LIV energy scale were imposed. Similarly, Ref. [225] failed to detect the LIV features from the H.E.S.S. observations of Mrk 501 during the strong flare recorded in June 2014: the non-detection of energy-dependent time delays and the absence of deviations between the measured spectrum and that of a supposed power-law intrinsic spectrum with standard EBL attenuation were used independently to derive strong constraints on the energy scale of LIV. CTA and ASTRI will be characterized by greatly improved sensitivity beyond this threshold, and allow us to probe the viability of the LIV scenarios. An important point is that prolonged exposures allow to reach the highest TeV-band energies [223]: the detected energy limit is expected to increase linearly with the exposure for photons with $E > 10$ TeV. Moreover, TeV-band observations at high zenith angles (corresponding to large effective areas of the TeV-band instruments), could be particularly favorable for the Lorentz LIV studies [223].

The current rare detections of ultra-fast VHE variability leaves the duty cycle (DC: fraction of the net exposure during which the object showed such a variability) of these events unknown: a proper evaluation of the duty cycle require a dense and frequent monitoring of the most promising targets, along with the enhanced sensitivity of the next-

generation instruments. Nevertheless, the DC can test and constrain some of the proposed variability scenarios [226]. Moreover, the events detected so far occurred only during the VHE flaring states of the sources, whereas it would be important to clarify whether such phenomena can also occur in states of low/quiescent activity or in the MeV–GeV range (not reported for HBLs to date).

Similar to other types of high-energy astrophysical systems, magnetic field is important for HBL physics. While the polarimetric studies can directly probe the magnetic field morphology and evolution in these systems, they still have not been carried out in the $\gamma$-ray energy range. Since the latter is associated with the most energetic processes, $\gamma$-ray polarization can probe more energetic phenomena in more extreme physical environments than X-rays. Gamma-ray polarimetry can directly disentangle the radiation mechanisms in relativistic jets and probe the existence of the anticipated hadronic signatures (e.g., the detection of a the MeV-band polarization will be a direct indication that the proton-synchrotron emission). Although the current hadronic models require higher jet powers and face more challenges in explaining of the very fast variability than leptonic scenarios, they still provide us with an important and physically motivated alternative for interpreting the available observational data. Gamma-ray polarization is also expected in the framework of the SSC and EIC models, but the proton–synchrotron radiation should be significantly more polarized [7,227]. The much higher sensitivity and spectral resolution of CTA compared to the current IACTs and expected coverage of the entire VHE range to above 100 TeV facilitates a search for potential hadronic signatures in the TeV spectrum.

A combination of very high angular resolution of the VLBI observations and very high temporal resolution to be achieved with CTA in the VHE range is crucial, particularly in the framework of Global VLBI Alliance which is under development [5]. The latter will have the power to resolve the inner jet regions, explore the detailed properties and the evolution of the magnetic field, and discern the physical mechanisms responsible for the jet-launching, particle acceleration and energy generation via the fast and multi-frequency VLBI study through the total and polarized light which is generated in the vicinity of the central SMBHs. Combined with the VHE monitoring, this technique is anticipated to firmly and accurately identify the sites of the TeV-band emission and flaring mechanisms; determine the importance of the BH magnetosphere in the generation of fastly variable TeV emission.

As noted above, the $\gamma$-ray studies can contribute to solve different fundamental problems of the modern physics and cosmology. For example, the LIV studies are useful for making a progress in the understanding of the intrinsic $\gamma$-ray spectra and variability of HBLs. Consequently, they will allow us to explore the nature of space-time via the propagation of their VHE photons, impact on our knowledge about the EBL, intergalactic magnetic fields (IGMF), facilitate axion-like particle searches, etc. Note that a LIV search was performed for Mrk 501 by [225] using the *Fermi*-LAT and H.E.S.S. data collected on 23–24 June 2014 when the source was showing rapid $\gamma$-ray variability. Based on the non-detection of energy-dependent time delays, as well as the absence of deviations between the measured spectrum and that of a supposed power-law intrinsic spectrum with standard EBL attenuation (anticipated within the LIV theory), some constraints related to the LIV energy scale were imposed.

Investigating whether relativistic shocks, reconnection zones, MHD turbulence or shear boundaries provide the dominant energization site for ultrarelativistic particles in HBLs represents the major target for future theoretical studies and highest-level simulations. Indeed, a realistic study of the jet internal shocks requires a time-dependent evaluation of the time-delayed radiation fields from all jet regions.

**Funding:** This research was funded Shota Rustaveli National Science Foundation grant FR-21–307.

**Institutional Review Board Statement:** Not applicable.

**Informed Consent Statement:** Not applicable.

**Data Availability Statement:** Data presented in Figures 1 and 4 are available upon request from the author.

**Acknowledgments:** The author thanks Shota Rustaveli National Science Foundation and E. Kharadze National Astrophysical Observatory (Abastumani, Republic of Georgia) for the fundamental research grant FR-21–307, Patrizia Romano for proofreading the manuscript and reviewers for their fruitful suggestions.

**Conflicts of Interest:** The author declares no conflict of interest.

## Note

1   See http://tevcat.uchicago.edu (accessed on 25 May 2023).

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
