# Peer review of "Gamma-ray Emission and Variability Processes in High-Energy-Peaked BL Lacertae Objects"

_universe, doi:10.3390/universe9070344_

Round 1
Reviewer 1 Report
The author has made an excellent effort to review the high energy emission from HBL blazars. I recommend publishing of the manuscript after implementing the minor corrections as suggested in the attachment.

The manuscript requires minor edits related to language.
Reviewer 2 Report

This manuscript would benefit from an additional round of copy editing before publication. In my comments above, I have mentioned only some particularly unclear passages, as well as obvious typos.
Reviewer 3 Report
I have reviewed the manuscript titled "Gamma-Ray Emission and Variability Processes in High-Energy-Peaked BL Lacertae Objects". It is a well written article that covers most of the important aspects of the gamma-ray emission from HSP blazars. I only have one major comment and several minor comments below in order of appearance.
My major criticism is that the review ignores the recent X-ray polarization results on HSPs (and LSP see why below). I understand that the focus is the gamma-ray emission from HSP, however, the review touches on the important topics of particle acceleration and the single-zone versus multi-zone models. It also mentions the potential of MeV polarization to differentiate between leptonic and hadronic processes. Therefore, I think that the recent X-ray polarization results suggesting that particle acceleration is taking place in shocks in an energy stratified manner in HSPs (and the results on LSPs pointing to leptonic processes) should be included in the overall discussion for completeness.
Minor comments:
I suggest removing the acronym definitions from the abstract.
"SED" defined multiple times.
line 160: "The this process" --> "This process"
Line 167 "Fort"--> "For"
line 221 "adopted A one-zone SSC" and later in line 226 "with A one-zone SSC scenario,"
line 243: " Consequently,a formation of the" --> " Consequently, the formation of the"
line 290: "paticles" --> "particles"
line 338: "the the synchrotron"
line 352: " the the SSC scenario"
line 425: " 1ES 022+200" --> " 1ES 0229+200"
line 446: "content;[83]" --> "content; [83]"
line 471: the second bullet point is part of the first one. Those should be merged. To make it more clear, I suggest starting each bullet point with the model that it i describing. E.g., \bullet {\bf Proton synchrotron.} In the so-called synchrotron-proton blazar (SPB) model [87] a significant portion of...
line 557: " the deduced Jet powers" --> " the deduced jet powers"
The fact that jet power requirements in hadronic models is a problem appears a bit out of the blue here. I suggest giving a little bit more discussion in that problem earlier when first talking e.g., about the magnetic field requirements.
line 660: "there was by a compact" --> "there was a compact"
line: 722 " needs a further confirmation." --> " needs further confirmation."
line 724 "e.g." in other places in the text it is given as "e.g.,". I suggest choosing one style throughout the text.
line 731: "Sock fronts" --> "Shock fronts"
line 772: "of a jet matter" --> "of jet matter" ... " triggers" --> " trigger"
line 796-797 "where a jet blob containing the separate acceleration and emission zones is moving" --> "where a jet blob, containing separate acceleration and emission zones, is moving"
line 833: "Initially,the magnetic " --> "Initially, the magnetic "
line 868: "mildly-relaitvistic shocks," --> "mildly-relativistic shocks,"
line 873: " for th IC" --> " for the IC"
line 875: " forward-reverse shock shock structure" presumably this is: " forward-reverse shock structure"
line 937: "Te model" --> "The model "
line 970: " variability is is modeled" --> " variability is modeled"
line 1048: "from a large scale down to the small wavelengths" --> "from large down to small wavelengths"
line 1064: "Hence the name of the name of the mechanism" --> "Hence the name of the mechanism"
line 1096: " the there can be a combined acceleration" --> " there can be a combined acceleration"
line 1101: "in the on which" --> "on which"
line 1103: " by he scattering" --> " by the scattering"
line 1161: "respectively,)," --> "respectively),"
line 1162: " are much significant:" --> " are much more significant:" or " are significant:"
line 1214: " through tho bubble" --> " through the bubble"
line 1217: " formation of double bow" --> " formation of a double bow"
line 1272: " polarimetrical " I would suggest using "polarization"
line 1319: "considered as one the candidate class" --> "considered as one candidate class"
line 1367: " magnetosphere n the generation " --> " magnetosphere in the generation "
line 1369 - 1374: this sentence is particularly hard to read and understand. I suggest rephrasing and breaking it to multiple sentences.
line 1374: " by [181] by using" --> " by [181] using"
line 1375: "showing a rapid γ-ray variability" --> "showing rapid γ-ray variability"
line 1379: "was imposed." --> "were imposed."
I have included my English language comments to the list above.
Reviewer 4 Report
Dear author,
I have read with interest the review on "Gamma-ray emission and Variability processes in High-Energy-Peaked BL Lacertae objects". I consider the review to be in general well written and pretty complete, and I do not any major criticism, although some relatively major requests/comments.
Relatively major points:
- it seems a bit strange that the paper doesn't mention the results from IXPE. Although they are not directly on gamma-ray emission, they do change quite a bit our view of the modeling of the emission region, and thus have impact on the modeling.
- on the hadronic models: it is also seem strange to not mention at all the TXS 0506+056 event. It constraints (if the association between the photon and neutrinos is real) the models quite a bit. The current paragraph on hadronic models seems thus a bit outdated, with pretty old references.
- again on the instruments: it is a bit strange to mention SWGO, but not HAWC or LHAASO, that have observed HBLs (although admittedly didn't have any groundbreaking results. But still, their contribution to the field should be put forward).
- the model in Figure 1 has to be redone: the numerical oscillations are way too strong and should be fixed.
Minor points:
line 35 - "there is a sequence" seems a bit too strong considering that there is no consensus. Please expand on this point.
line 102 - "while protons do noT posses sufficient energies". This is not the only reason: the protons could exist, and with high energies, just not in a high enough number to dominate the electromagnetic emission over the photons.
line 132 - " the adiabatic expansion is neglected" is incorrect: the differential energy cannot have only an injection and cooling term, there has to be an adiabatic/escape term: otherwise there is no equilibrium, and the cooling break goes to gamma=1.
line 169: the condition on the infrared luminosity is very unclear: if there is no external photon field, how could it depend on the black hole mass? Please expand and add citations.
line 229: none of the TeV HBLs is bright in the LAT band. Please reformulate.
line 256: the emission will contain -> the emission will NOT contain
line 358: TeV gamma-rays are not upscattered, please check the sentence.
line 460: what author calls "purely hadronic model" is basically not existent in the literature. I suggest the author to keep the leptonic/hadronic classification on the basis of the high-energy SED component: the fact that synchrotron electron is at the origin of the low-energy emission is taken as a matter of fact by everyone. Hadronic model is then a model in which the gamma-ray emission doesn't need any primary electron emission (i.e. SSC or EIC). A Lepto-hadronic models is then a model in which the gamma-ray emission is due to both SSC/EIC by primaries and hadronic emission (at some variable ratio)
line 503: beware that the Bethe-Heitler pair production is NOT associated to the neutrinos. Please modify this paragraph (Also, the Bethe-Heitler emission is known since well before 2015.. please reformulate the sentence before the citation).
line 529: by using the reasonable values? Which one?
line 549: 10^46 is not super Eddington. It depends on the black hole mass.
line 589: there is a contradiction in this sentence: if there is efficient gamma-gamma pair production, then the TeV band is opaque, so it is difficult to understand the "peak at the TeV frequencies". Please expand
line 595: accretion disks (of SMBH) do not emit MeV photons. Please correct
line 652: the sentence on the BLR is incorrect: the fact that matter flows in the vicinity of the jet in BL Lacs, doesn't mean that these clouds are ionized and emit photons. There is certainly matter there, but not illuminated by the disk.
line 653: the Breit-Wheeler pair production is unknown to me. Please add a reference.
line 728: please not that the majority of the claims on QPO lack any meaningful treatment of trial corrections. Some QPOs exist (and 1553 is not the most significant of them), but several others are certainly false positive. See Ren, H. et al. 2023 (A&A 672, 86).
line 766: please introduce V_sh
line 786: please define the shock compression ratio
line 824: please avoid the use of etc
line 842: how can a shock be superluminal?
line 875: what is a Mach disk?
Figure 4: Add citation.
Line 1243: it is unclear why this part on the log normality is here. It is still in the 3.4 section on jet-star interactions. Shouldn't this be a separate section?
line 1279: ASTRI is not within the framework of CTAO, it is an independent instrument.
line 1282: the 10-100 TeV band is not unexplored, see above. LHAASO and HAWC are taking data.
line 1285: please specify if missions such as amigo or e-astrogam are proposals, or funded projects.
Dear author,
Please note that, on the other hand, there are several English mistakes that make the paper difficult to read, especially in the second half. Please review carefully the paper before resubmitting. I provide here a list of corrections but it is certainly not complete:
line 50: astrong -> a strong
line 71: the origin OF
line 106: of the external -> of external
line 128: OT the blob?
line 133: is described -> described
line 158: "as strictly" is incomplete, it requires a comparison with something that is not written
line 160: The this
line 176: can to be
line 181: up TO
line 229: with the steep -> with steep
line 255: even in the case -> even if
line 287: the electrons repeated twice
line 290: particles
line 326: the electron population -> an electron population
line 330: with which the one observes -> rephrase
line 332: "relies"? unclear, please reformulate
line 346: injected IN? a pipe?
line 355: X-rays do not "collide", they scatter
line 393: problematic TO constrain THE free parameters
line 404: eletron
line 407: higher -> high (comparison terms need to be followed by the compared term)
line 414: TO reproduce
line 416: existence of multiple
line 418: is the subject of a rapidly
line 445: is contains -> contains
line 480 : the significant -> significant
line 515: of of
line 520: anticipated? Please change verb here
line 545: approach -> approaching
line 576: later -> latter
line 577: the area -> an area
line 626: cab -> can
line 660: there was by?
line 774: eternal perturbation? unclear
line 783: a an
line 792: they -> their
line 802: Within -> within
line 821: becomes decreases -> decreases
line 833: thoughT
line 838: can BE triggered
line 845: when THEY undergo
line 850: (if I understood correctly) in order a shock-drift acceleration to be -> FOR a shock-drift acceleration to be
line 850: In contrast -> in contrast
line 873:thE
line 968: the functions of?
line 970: is is
line 1059: towards both
line 1064: the name is duplicated
line 1094: may BE present
line 1101: in the on which -> at which
line 1128: have have
line 1165: when the particle energy becomes.
line 1185: small then -> smaller than
line 1214: through tho
line 1217: of A double
line 1302: to reach
line 1303: offers enhanced capabilities
line 1330: prolonged allow? unclear
line 1360: later -> latter
line 1367: n -> and?
line 1369: to solving the
Round 2
Reviewer 4 Report
Dear author,
thank you very much for the new version of the paper; I am happy to recommend it for publication.
I don't have any additional correction to report; but I suggest the paper to be sent to language editing before printing.